# KBLaM: Knowledge Base augmented Language Model

**Xi Wang**[†][*]
Johns Hopkins University
xwang457@cs.jhu.edu

**Taketomo Isazawa**[*]
Microsoft Research
t-isazawat@microsoft.com

**Liana Mikaelyan**
Microsoft
lmikaelyan@microsoft.com

**James Hensman**
Microsoft Research
jameshensman@microsoft.com

## Abstract

In this paper, we propose **K**nowledge **B**ase augmented **La**nguage **M**odel (KBLaM), a new method for augmenting Large Language Models (LLMs) with external knowledge. KBLaM works with a knowledge base (KB) constructed from a corpus of documents, transforming each piece of knowledge in the KB into continuous key-value vector pairs via pre-trained sentence encoders with linear adapters and integrating them into pre-trained LLMs via a specialized rectangular attention mechanism. Unlike Retrieval-Augmented Generation, KBLaM eliminates external retrieval modules, and unlike in-context learning, its computational overhead scales linearly with KB size rather than quadratically. Our approach enables integrating a large KB of more than 10K triples into an 8B pre-trained LLM of only 8K context window on one single A100 80GB GPU and allows for dynamic updates without model fine-tuning or retraining. Experiments demonstrate KBLaM's effectiveness in various tasks, including question-answering and open-ended reasoning, while providing interpretable insights into its use of the augmented knowledge. Code and datasets are available at https://github.com/microsoft/KBLaM/

## 1 Introduction

Large language models (LLMs) have demonstrated impressive knowledge and reasoning capabilities. However, in many scenarios, users need to augment LLMs with external knowledge, particularly when concept definitions differ from or extend beyond the information stored in the LLM's parameters. One straightforward approach is to perform supervised fine-tuning on the external corpus (Hu et al., 2021; Liu et al., 2024c) to alter the parameter weights. However, fine-tuning can be inefficient as the weights need to be updated each time knowledge is updated. Additionally, fine-tuning can cause catastrophic forgetting, i.e. degrading performance on general tasks. As such, alternative methods that do not modify LLM's weights have gained more and more popularity.

One such alternative approach is Retrieval Augmentation Generation (RAG, Lewis et al., 2020). RAG assumes external knowledge is stored as unstructured documents, then given an input prompt, RAG first uses a retriever module to extract relevant corpus chunks from the documents. These extractions are then concatenated to the input question as a prompt and fed into the LLM (Fig. 1 top). This approach allows RAG to utilize knowledge from extensive documents while maintaining a small context size, overcoming the context limitations of many early-generation LLMs.

Recently, with the emergence of long-context language models such as GPT4 (Achiam et al., 2023) and Gemini (Team et al., 2023), it has become possible to directly put the entire external corpus in the context, eliminating the need for the retriever for document selection, simplifying the pipeline. We refer to this approach as *in-context learning* (Fig. 1 middle). Several recent works (Lee et al., 2024; Li et al., 2024a; Yu et al., 2024) have explored whether long-context models can replace RAG in long-context reasoning tasks. In particular, Lee et al. (2024) argues that in-context learning has

---

[*]Equal contribution. † Work done during an internship at Microsoft Research.

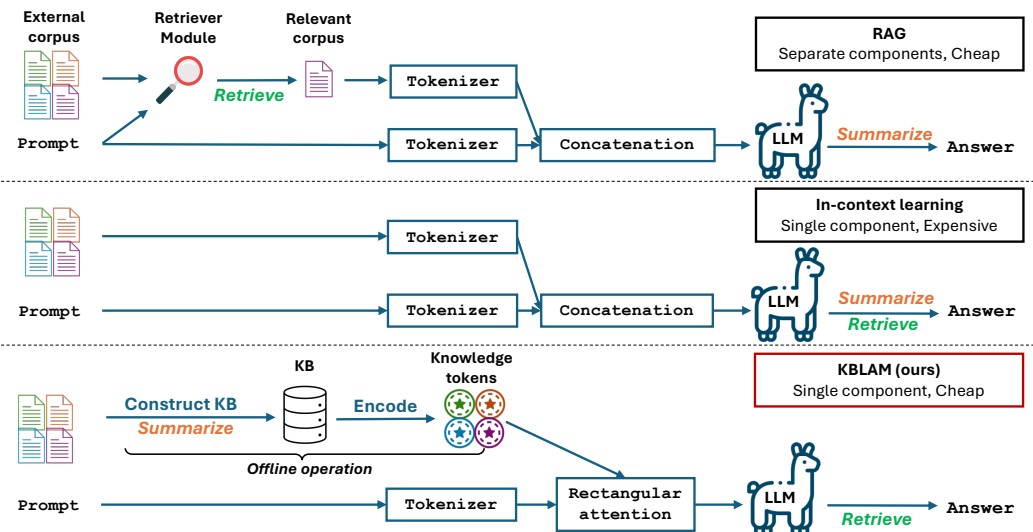

Figure 1: **Overview of the KBLAM pipeline and comparison with existing approaches.** KBLAM augments knowledge into a pre-trained LLM in the form of knowledge tokens, a set of continuous key-value vectors, using a modified rectangular attention structure. Unlike RAG, KBLAM does not rely on separate retriever module at inference time and unlike in-context learning, KBLAM's computation and memory overhead scales *linearly* rather than quadratically with the size of the KB.

potential advantages, such as better compatibility with chain-of-thought reasoning and few-shot demonstrations. However, the time and memory complexities of in-context learning are quadratic with the context length. Additionally, the dependencies between tokens in the context introduce difficulties in interpreting the attention matrix and dynamically updating the KV cache.

In this paper, we propose a new regime for augmenting pre-trained LLMs with external knowledge: **K**nowledge **B**ase augmented **La**nguage **M**odel (KBLAM, Fig. 1 bottom). We consider a setting where the *unstructured* external corpus is transformed into a *structured* knowledge base (KB) through existing tools, which summarizes the critical information in the data and generates knowledge triples (Eq. (1)) containing an entity name (`<name>`), a property (`<property>`), and a value (`<value>`). Then, KBLAM's design fully leverages the structure of the KB, achieving efficient integration of external knowledge into LLMs (Sec. 4).

First, KBLAM maps each triple into a *fixed-length* key-value vector pair, referred to as a *knowledge token*, which has sizes identical to the KV cache of a single token and can be seamlessly incorporated into each attention layer of the LLM. In particular, KBLAM encodes from `<name>` and `<property>` into a key vector, which serves as an identifier, mimicking the key embedding of a token; `<value>` into a value vector, which provides the actual content, similar to the value embedding of a token.

Then, KBLAM uses the structure between triples to augment knowledge tokens into the LLM's attention in a scalable and dynamic way using a simple rectangular attention structure: Triples with different `<name>` and `<property>` can be considered to represent independent pieces of information, therefore knowledge token from each triple can be encoded and injected into pre-trained LLM independently. This allows the KBLAM's complexity (memory and time) to grow linearly with respect to the number of triples, unlike in-context learning's quadratically growing overhead, giving KBLAM much better scalability. Additionally, the independence allows us to update/remove/add a triple by only modifying its corresponding single knowledge token without any further changes, which is not achievable in, e.g. standard KV cache mechanism. Perhaps more importantly, the attention matrix under this design is highly interpretable. As we show in Fig. 4, the model's use of the knowledge tokens can be directly inspected through the attention score.

Lastly, we show that the linear adapters can be learned using instruction tuning on purely synthetic data (Sec. 5) while being able to generalizes to real data. Different from supervised fine-tuning, which aims to memorize knowledge into model weights, the learning process of KBLAM aims at

finding a projection between the pre-trained sentence encoder space and the LLM's embedding space, which motiviates us the train KBLAM with fully synthetic data since the exact knowledge in the training samples is not important.

Compared with RAG, KBLAM incorporates all external information in the context and answers input prompts in an end-to-end way without utilizing additional external modules, similarly to in-context learning. Indeed, KBLAM also shows empirical performance comparable to in-context learning but with significantly better scalability to more than 10K triples. In addition, through instruction tuning, KBLAM can learn to refuse to provide an answer if the information required to answer the question is not present in the KB, improving the reliability of the model and reducing hallucinations.

In summary, KBLAM is a novel method for augmenting pre-trained LLMs with external knowledge bases (KBs). We demonstrate that KB triples can be efficiently encoded into "knowledge tokens" - continuous key-value vector pairs equivalent in size to one LLM token - using a pre-trained sentence encoder with a learned linear adapter. These tokens are incorporated into the LLM through a modified attention structure, enabling linear scaling with KB size and allowing dynamic knowledge updates without model fine-tuning. The adapters are trained end-to-end using instruction tuning on synthetic data. Lastly, we release our training and evaluation KBs, which can help future research in augmenting LLM with KB, as well as in other topics such as long-context language models, hallucination detection/reduction, and structured attention.

## 2 RELATED WORK[1]

**Retrieval augmented generation (RAG)** RAG (Lewis et al., 2020) is one of the most successful approaches for augmenting external knowledge into LLMs, and many approaches have been proposed to perform RAG, such as similarity search and text2Sql (Qin et al., 2022). KBLAM can also be understood as the model doing RAG in an implicit way: given input prompts, the rectangular attention compares the sequence's queries with knowledge tokens' keys, then the weighted average is taken of all knowledge tokens' values based on query-key similarity, similarly to a soft retrieval process. The averaged value embedding is then added to the hidden state, augmenting next-token generation with information from the KB. However, KBLAM differs from RAG in that it does not rely on a separate retrieval module. Instead, everything is carried out by attention in an end-to-end fashion.

**Structured attention** KBLAM also shares similarities with works that use structured attention masks instead of the standard lower-triangular causal attention mask (Ratner et al., 2022; Cai et al., 2023; Merth et al., 2024). These works utilize independence assumptions in the context, e.g. between different randomly-chosen sets of few-shot examples (Ratner et al., 2022) or documents from different sources (Cai et al., 2023; Merth et al., 2024), and let independent contexts not attend over each other, which reduces the overhead of attention operations. In KBLAM, we also utilize similar independence assumption among inputs: We assume independence between different KB triples, which provides KBLAM with scalability and interpretability.

**Key-value (KV) cache mechanism** Given a context string, the KV cache mechanism (Pope et al., 2023) caches the key and value embeddings at each layer, also known as the prefill stage. With KV cache, the complexity of generating new tokens conditioned on the context decreases from quadratic in context length to linear. KBLAM's encoded knowledge tokens work in a way similar to KV cache, but we have the key and value acquired from an external encoder rather than through self-attention process. Additionally, if the context is modified, the standard KV cache requires re-computing the whole KV cache due to the causal attention mask, whereas in KBLAM only the corresponding knowledge token need be updated as the knowledge tokens do not attend to each other.

## 3 BACKGROUND

**Knowledge base in the form of triples** In this paper, we assume that the external knowledge is represented as a structured knowledge base (KB) obtained from unstructured text using, in particular, a KB is composed of triples of format

$$\{(\texttt{<name>}_m; \texttt{<property>}_m; \texttt{<value>}_m)\}_{m=1}^M. \tag{1}$$

In the rest of the paper, we will refer to a set of triples of the format Eq. (1) as a KB and each triple inside it as a *knowledge triple*. Examples of such KB is shown in Table. 1 and 2 in Appendix E.

---

[1] We provide more extensive related work discussion in Appendix A

The KB construction process summarizes information from multiple documents and organizes it in a structured way. Importantly, the structured nature of the KB enables us to augment LLM with information from documents in an efficient way, which we discuss in the next section.

It is also worth noting that KB construction is *not* the focus of our paper: KBLAM works on KB generated from corpus using *existing tools*. In the paper, we mainly focus on two KBs: A synthetic KB generated via GPT, and a real KB constructed from the Enron email (Klimt & Yang, 2004) dataset. Details of the two KBs will be discussed in detail in later sections.

**Self-attention layer** A decoder-based transformer is mostly composed of multiple self-attention layers (Vaswani, 2017). Consider a transformer of $L$ layers. Each layer has three projection heads: $\boldsymbol{W}_Q^l \in \mathbb{R}^{D \times D}, \boldsymbol{W}_K^l \in \mathbb{R}^{D \times D}, \boldsymbol{W}_V^l \in \mathbb{R}^{D \times D}, l \in \{1, \ldots, L\}$, where $D$ denotes *embedding dimension*. Given a user prompt (e.g. a question) of $N$ tokens, each attention layer takes a sequence of $N$ token embeddings $\boldsymbol{x}^l = [\boldsymbol{x}_1^l, \ldots, \boldsymbol{x}_n^l, \ldots \boldsymbol{x}_N^l]^\top \in \mathbb{R}^{N \times D}$ as input, then for the $n$th token embedding, it is first transformed by the projection heads into three vectors

$$\boldsymbol{q}_n^l = \boldsymbol{W}_Q^l \boldsymbol{x}_n^l, \qquad \boldsymbol{k}_n^l = \boldsymbol{W}_K^l \boldsymbol{x}_n^l, \qquad \boldsymbol{v}_n^l = \boldsymbol{W}_V^l \boldsymbol{x}_n^l, \qquad (2)$$

where we refer to $\boldsymbol{q}_n^l$, $\boldsymbol{k}_n^l$, and $\boldsymbol{v}_n^l$ as the $n$th token's query, key, and value embedding at the $l$th layer respectively. Note that in practice, each attention layer often has multiple sets of attention heads but here we assume only a single head for notation simplicity.

The output of the attention layer, $\boldsymbol{y}^l = [\boldsymbol{y}_1^l, \ldots, \boldsymbol{y}_n^l, \ldots, \boldsymbol{y}_N^l]^\top \in \mathbb{R}^{N \times D}$, is computed as

$$\boldsymbol{y}_n^l = \frac{\sum_{i=1}^n \exp(w_{n,i}) \boldsymbol{v}_i^l}{\sum_{i=1}^n \exp(w_{n,i})}, \text{ where } w_{n,i} = \langle \boldsymbol{q}_n^l, \boldsymbol{k}_i^l \rangle / \sqrt{D} \qquad (3)$$

where $\langle \cdot, \cdot \rangle$ denotes the inner product of two vectors. After $\boldsymbol{y}_n^l$ is acquired, it is often fed into a feedforward network (FFN) with output dimension of $D$ for further transformation.

A standard implementation of self-attention would have a time complexity of $\mathcal{O}(N^2 D)$ and memory complexity of $\mathcal{O}(N^2)$ for computing and storing all $w_{n,i}$. Additionally, the FFN would also introduce a significant computation overhead of magnitude $\mathcal{O}(ND^2)$. Due to these factors, as the sequence gets longer, self-attention faces high memory consumption and slow computation.

## 4 AUGMENTING LLM WITH THE KB

In this section, we discuss how KBLAM augments an LLM with a KB. The process has two steps (visualized in Fig. 2): 1. We convert each triple in the KB from the string form into a continuous key-value vector pair, referred to as a *knowledge token*, through a pre-trained sentence encoder followed by linear adapters; 2. All knowledge tokens are then injected into each attention layer of an LLM through a *rectangular attention* structure.

**Knowledge tokens** Provided a KB in the format of Eq. (1), for each triple, we first adopt a *pre-trained* sentence encoder model, denoted as $f(\cdot)$ which converts a string into a $P$-dimension continuous embedding vector. Through the encoder, we convert each triple into a base key embedding and a base value embedding. In particular, for the $m$th triple, we have:

$$\mathbf{k}_m = f(\texttt{The <property>}_m \texttt{ of <name>}_m) \in \mathbb{R}^P, \quad \mathbf{v}_m = f(\texttt{<value>}_m) \in \mathbb{R}^P. \qquad (4)$$

Next, we introduce a linear key and a linear value adapter

$$\tilde{\boldsymbol{W}}_K \in \mathbb{R}^{L \times D \times P}, \qquad\qquad \tilde{\boldsymbol{W}}_V \in \mathbb{R}^{L \times D \times P}, \qquad (5)$$

where $L$ denotes the number of attention layers in a model. With the adapters, we map $\mathbf{k}_m$ and $\mathbf{v}_m$ from the sentence encoder's space to LLM's key and value embedding space at each attention layer. In particular, for the $m$th knowledge triple, we transform its base key and value embedding into

$$\tilde{\boldsymbol{k}}_m = [\tilde{\boldsymbol{k}}_m^1, \ldots, \tilde{\boldsymbol{k}}_m^l, \ldots, \tilde{\boldsymbol{k}}_m^L]^\top = \tilde{\boldsymbol{W}}_K \mathbf{k}_m \in \mathbb{R}^{L \times D}, \qquad (6)$$

$$\tilde{\boldsymbol{v}}_m = [\tilde{\boldsymbol{v}}_m^1, \ldots, \tilde{\boldsymbol{v}}_m^l, \ldots, \tilde{\boldsymbol{v}}_m^L]^\top = \tilde{\boldsymbol{W}}_V \mathbf{v}_m \in \mathbb{R}^{L \times D}. \qquad (7)$$

Since each $\tilde{\boldsymbol{k}}_m^l$ and $\tilde{\boldsymbol{v}}_m^l$ share the same size with LLM's key and value embedding (i.e. $\boldsymbol{k}_n^l$ and $\boldsymbol{v}_n^l$ in Eq. (2)), they can be directly incorporated into attention's computation. Therefore, the encoding

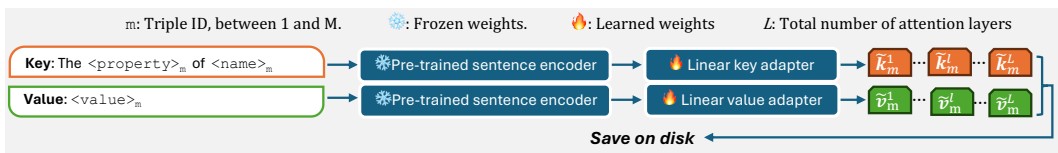

(a) **Step 1. KB encoding** For each triple from a KB (Eq. (1)) of $M$ triples, we first construct a key and value string (left most white boxes). Then both strings are processed by a pre-trained sentence encoder followed by a learned linear adapter (middle two blue boxes). The acquired knowledge tokens (Eqs. (6) and (7), right-most small boxes) are then stored on disk for later usage. Note that all $\tilde{\boldsymbol{k}}_m^l$s and $\tilde{\boldsymbol{v}}_m^l$s have a fixed dimension of $D$, i.e. identical to that of the key and value embeddings of a token, regardless of input length.

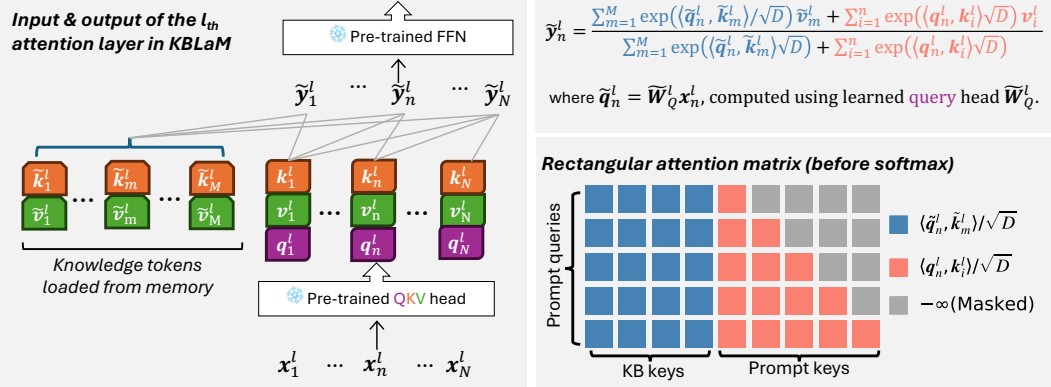

(b) **Step 2. Augmenting knowledge tokens into attention** For layer $l$, given an sequence of $N$ $D$-dimensional embeddings from the prompt $(\boldsymbol{x}_1^l, \ldots, \boldsymbol{x}_N^l$, e.g. a question about the KB), augmented with $M$ knowledge tokens as context $(\{(\tilde{\boldsymbol{k}}_m^l, \tilde{\boldsymbol{v}}_m^l)\}_{m=1}^M)$, KBLAM's attention outputs $N$ embedding vectors with each element $\tilde{\boldsymbol{y}}_n^l \in \mathbb{R}^D$ computed via Eq. (9) (top right panel) under a rectangular attention matrix (bottom right panel): Blue regions show the extra components introduced by KBLAM whereas the red parts are from standard self-attention. Note that the input sizes of the FFN stay unchanged, the only additional overhead introduced by KBLAM comes from the blue parts in the equation and the matrix, which scales as $\mathcal{O}(M)$.

Figure 2: Overview of KBLAM's KB augmentation process

process is equivalent to converting a knowledge triple from a string of multiple tokens into a special token, whose key and value embeddings at each layer are generated through an encoder, rather than through self-attention. As such we refer to $(\tilde{\boldsymbol{k}}_m, \tilde{\boldsymbol{v}}_m)$ pair as a *knowledge token*.

We visualize the process in Fig. 2a. This encoding process is applied to all triples in the KB, which transforms the information from a KB into a collection of knowledge tokens

$$\{(\texttt{<name>}_m, \texttt{<property>}_m, \texttt{<value>}_m)\}_{m=1}^M \xrightarrow{\text{Encode}} \{(\tilde{\boldsymbol{k}}_m, \tilde{\boldsymbol{v}}_m)\}_{m=1}^M. \tag{8}$$

**Rectangular Attention: Injecting knowledge token into prompt tokens** After converting a KB into a collection of knowledge tokens we inject the information of the KB into the each attention layer of a pre-trained LLM through a modified attention structure[2]. In particular, consider an input prompt of $N$ tokens, the $l$th attention layer takes in $\boldsymbol{x}^l = [\boldsymbol{x}_1^l, \ldots, \boldsymbol{x}_n^l, \ldots, \boldsymbol{x}_N^l]^\top$ and the $l$th dimension of the knowledge tokens $\{(\tilde{\boldsymbol{k}}_m^l, \tilde{\boldsymbol{v}}_m^l)\}_{m=1}^M$, the output of each attention layer, $\tilde{\boldsymbol{y}}^l = [\tilde{\boldsymbol{y}}_1^l, \ldots, \tilde{\boldsymbol{y}}_n^l, \ldots, \tilde{\boldsymbol{y}}_N^l]^\top \in \mathbb{R}^{N \times D}$, is given by

$$\tilde{\boldsymbol{y}}_n^l = \frac{\sum_{m=1}^M \exp\left(\tilde{w}_{n,m}^l\right) \tilde{\boldsymbol{v}}_m^l + \sum_{i=1}^n \exp\left(w_{n,i}^l\right) \boldsymbol{v}_i^l}{\sum_{m=1}^M \exp\left(\tilde{w}_{n,m}^l\right) + \sum_{i=1}^n \exp\left(w_{n,i}^l\right)}, \tag{9}$$

where

$$\tilde{w}_{n,m}^l = \langle \tilde{\boldsymbol{q}}_n^l, \tilde{\boldsymbol{k}}_m^l \rangle / \sqrt{D}, \ \tilde{\boldsymbol{q}}_n^l = \tilde{\boldsymbol{W}}_Q^l \boldsymbol{x}_n, \qquad w_{n,i}^l = \langle \boldsymbol{q}_n^l, \boldsymbol{k}_i^l \rangle / \sqrt{D}, \tag{10}$$

and $\tilde{\boldsymbol{W}}_Q^l \in \mathbb{R}^{D \times D}$ is a *learnable* query head initialized from $\boldsymbol{W}_Q^l$.

---

[2]We use symbols with a tilde to denote additional attention components introduced by KBLAM.

We name the attention structure of Eq. (3) *rectangular attention* as the attention matrix is of rectangular shape (bottom-right panel in Fig. 2b). The idea is that tokens in the prompt part can attend to, in addition to all previous tokens in the prompt, all knowledge tokens. On the other hand, knowledge tokens cannot attend to each other, i.e. they do not have self-attention nor query embeddings. This design gives an attention matrix of rectangular shape of size $(M+N) \times N$, where $M$ is the number of triples in the KB. Additionally, if knowledge tokens are not introduced, i.e. $M = 0$, Eq. (9) falls back to Eq. (3) and we recover the pre-trained LLM.

The memory and time complexity of the rectangular attention is $\mathcal{O}((M+N)N)$ and $\mathcal{O}((M+N)ND)$ respectively. In real settings, it is likely that $M >> N$, i.e. the number of external knowledge items is far larger than the size of the prompt/question, then the overhead would grow only linearly with $M$ (instead of quadratically), which allows KBLAM to scale up to very large values of $M$, enabling putting a large amount of information in the context. Also, note that the output sequence length of rectangular attention does not vary with the number of knowledge tokens, therefore the overhead for the intermediate FFN modules stays unchanged regardless of $M$.

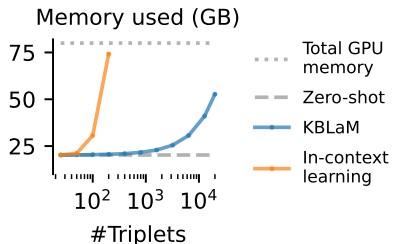

Memory used (GB)

#Triplets

Figure 3: Memory overhead of different methods. Given a KB of $M$ triples (with each triple $K$-token long on average). In-context learning's memory scales with $(KM)^2$, whereas KBLAM's memory scales with $M$.

Lastly, it can be seen from Eq. (3) that changing the order of the knowledge tokens does not affect the output, therefore KBLAM does not suffer from the positional bias issue of in-context learning (Liu et al., 2024b). Additionally, since all knowledge tokens are encoded independently, when the KB is updated, e.g. new knowledge triples are created or a triple is updated, one could simply generate a new knowledge token with the encoder or update the corresponding knowledge tokens value. This highlights the difference between KBLAM and the standard KV cache mechanism in LLM, which needs re-computation when the cached content is modified in any way.

Note that KBLAM's rectangular attention *differs* from cross-attention (Vaswani, 2017; Chen et al., 2021), which operates on a source sequence with key and value embeddings and a target sequence with only query embeddings, where tokens within each sequence do not attend to one another. In our setting, if we consider the KB as the source and the input prompt as the target, cross-attention corresponds to retaining only the summations over $M$ in Eq. (9). However, KBLAM's attention also contains self-attention over the prompt tokens, given by the summations over $n$ in Eq. (9).

**KB length generalization through attention score scaling**  Notice that the part of the attention output corresponding to the KB would scale with $M$. In particular, assume $\tilde{w}_{n,m}^l, m \in \{1, \ldots, M\}$ are all of similar magnitude, then as $M$ grows, eventually the second term (the prompt/questions part) in both denominator and numerator of Eq. (9) will be overwhelmed by the first term (the KB part), and the information from the prompt/question will be lost. To resolve this, during *inference*, we add a constant shift to the KB part's unnormalized attention score

$$\tilde{w}_{n,m}^l = \log C - \log M + \langle \tilde{\boldsymbol{q}}_n^l, \tilde{\boldsymbol{k}}_m^l \rangle / \sqrt{D}, \tag{11}$$

where $C \in \mathbb{R}^+$ is a hyperparameter. This is equivalent to multiplying the summations over $M$ by $\frac{C}{M}$, such that the total contribution of the KB part stays roughly unchanged as $M$ scales. In our experiments, we fix $C$ as 100, which is the largest number of triples (i.e. $M$) seen at training time.

## 5 KB INSTRUCTION TUNING

The learnable parameters in KBLAM comprise the weights for the weights for the linear adapters/heads $\theta = \left\{ \tilde{\boldsymbol{W}}_K, \tilde{\boldsymbol{W}}_V, \{\tilde{\boldsymbol{W}}_Q^l\}_{l=1}^L \right\}$, Inspired by recent works in multi-modal language models (Liu et al., 2023), we use instruction tuning for parameter learning. Specifically, given a KB, we generate question-answer pairs about the KB (denoted as $Q$ and $A$ respectively) using formatted strings or GPT, then we optimize $\theta$ using:

$$\max_\theta \log p_{\theta,\phi}(A \mid Q, \text{KB}), \tag{12}$$

where $\phi$ denotes parameters of the pre-trained LLM, encompassing QKV heads, FFN parameters and embedding/outpu layer weights. Notably, KB instruction tuning preserves the base LLM's reasoning

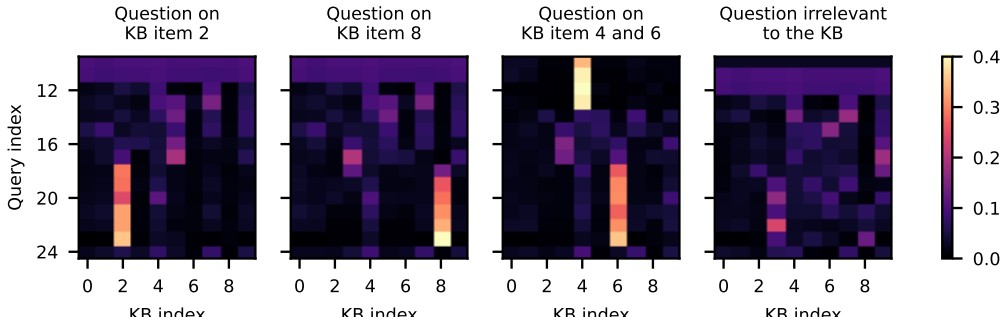

Figure 4: **KBLAM's attention matrix is interpretable.** Consider a toy KB of 10 triples, the heatmap shows the attention weights under different questions. We select the 15th attention layer and visualize the post-softmax attention score averaged over all attention heads. The x-axis only shows the keys corresponding the KB part and the y-axis is aligned by the end of each question's query.

abilities by avoiding fine-tuning the pre-trained LLM itself. Moreover, learning only the linear adapters minimizes the risk of memorizing training data, a known issue in LLM fine-tuning (Zeng et al., 2023; Mireshghallah et al., 2022). To further mitigate this risk, we conducted instruction tuning using a *fully synthetic* KB generated by GPT. This approach is motivated by the understanding that the instruction tuning process aims *not* to memorize specific information, but to learn a projection from the pre-trained sentence encoder space to the LLM's semantic space. Consequently, the exact content of the KB is less critical, provided the training data encompasses a diverse range of text.

**Generation of synthetic KB** To synthesize a KB, we first use GPT to generate 50 `<name>`s based on combinations of 30 object types (e.g. restaurant name, software tool) and 30 idea types (e.g. natural phenomena, famous landmarks). Then for each name, we prompt GPT to generate the `<value>` of three `<property>`s: "description", "objectives", and "purpose" for each name, in the same conversation context. Crucially, we instruct GPT to generate `<value>`s *uncorrelated* with names, ensuring the information comes from the KB rather than the LLM's predictive ability. This yields 45K names and a KB of approximately 135K triples. We provide example triples in Table. 1 in Appendix. E, the full list of types and prompts in Appendix. G.1 and release the resulting synthetic KB dataset for research purposes.

**Generation of instructions** The construction of the instruction tuning dataset is crucial, as it reflects our intended use of the KB and the desired behavior of KBLAM under various scenarios. We considered the following types of instructions (examples shown in Fig. 12 in Appendix. F):

- **Simple Q&A** about a single `<name>` and `<property>` with `<value>` as the answer.
- **Multi-entities Q&A** involving multiple `<name>`s and `<property>`s, with their corresponding `<value>`s as answers.
- **Open-ended Q&A** similar to simple and multi-entities Q&A, but including an additional open-ended question component that requires open-ended reasoning about `<value>`. Reference answers for these questions are generated by GPT. The exact prompt is detailed in Appendix. G.2.
- **Unanswerable questions** irrelevant to any `<name>` or `<property>` in the KB. For such cases, the standard response is "Sorry, I cannot find relevant information in the KB."

For simple, multi-entities, and unanswerable questions, both questions and answers are generated using formatted strings, with templates of various styles such as "Can you inform me..." or "What is the...". A comprehensive list of these templates is provided in Appendix. G.5.

## 6 EXPERIMENTS

In this section, we perform empirical evaluation for KBLAM. We begin by showing that KBLAM's attention matrix, after instruction tuning, shows interpretable patterns and works as an accurate retriever. We then show that KBLAM can answer questions with performance comparable to in-context learning, but with much lower memory cost, and can scale to 10K triples with little performance degradation. Lastly, we show that for questions with no answer in the KB, the model can refuse to answer questions, with "over-refusal" behavior occurring later than in-context learning. Lastly, we conduct ablation studies to understand the design choices of KBLAM.

## 6.1 EXPERIMENT SETTING

**Model specification** For all experiments, we use the instruction fine-tuned version of Llama3 8B (Dubey et al., 2024) as the backbone LLM, and OpenAI's ada-002 sentence embedding model ($P = 1536$) as the pre-trained encoder for computing base key and value embedding (Eq. (5)).

**Optimization setting** We initialize the key and value adapters: $\tilde{\boldsymbol{W}}_K$ and $\tilde{\boldsymbol{W}}_V$ randomly, and we initialize additional query heads $\tilde{\boldsymbol{W}}_Q^l$ at each layer from the pre-trained weights $\boldsymbol{W}_Q^l$. Optimization is conducted using AdamW (Loshchilov, 2017) with a step size of $5 \times 10^{-4}$ and a cosine learning rate decay to $5 \times 10^{-6}$ for 20K iterations. Each iteration uses a mini-batch of 400 Q&A pairs, composed of 20 micro-batches of 20 samples. The instruction tuning is performed on a single 80GB A100 GPU under bfloat16 without any parameter-efficient tuning methods.

**Construction of the training batches** We train KBLAM using instruction tuning, where each training sample consists of a KB, a question, and an answer (Eq. (12)). These samples are derived from the first 120K triples of the synthetic KB described in Sec. 5. To construct each training sample, we perform the following procedure: We randomly select a subset of 10 to 100 triples from the synthetic KB to form a sample-specific KB. Depending on the instruction type (simple, multi-entities, open-ended, or unanswerable Q&A), we designate one, multiple, or no triples as the *relevant* triple(s), with the rest serving as *distractors*. We generate a question based on the instruction template and the relevant triple(s). In each batch of 20 micro-batches, we include 2 micro-batches of unanswerable Q&A pairs and 6 micro-batches each for the other three Q&A types. During training, we find limiting the KB size crucial for successful convergence. To further optimize training efficiency, we pre-compute the base key and value vectors (Eq. (5)) offline for all triples in the synthetic KB.

**Evaluation dataset** For evaluation, we considered the following two KB datasets[3]

- **Synthetic data** The validation set of the synthetic KB, i.e. the 15000 triples not used for training.
- **Enron** A KB constructed from the Enron (Klimt & Yang, 2004) dataset, an open-sourced corporate email dataset. Following (Wu et al., 2024), we constructed a KB of triples from the Enron emails data by fine-tuning a small language model, which we then cluster to remove duplicates (Winn et al., 2018; 2021).

For each validation dataset, we consider `<property>` in "description", "purpose" and "objectives".

**Baseline** We considered the following two methods as baselines

- **In-context learning** Flatten all triples in a KB as strings, and attach it in front of the prompt. In-context learning's memory overhead grows quadratically with the number of triples. Notice that we can only experiment with a maximum of 200 triples due to memory constraints.
- **Zero-shot learning** Directly ask the LLM a particular question and provide no additional context, using the LLM's internal knowledge for answering the questions.

**Evaluation setting** For all evaluations, we repeat the experiments with 5 random seeds, each run containing 100 test samples, i.e. we randomly generate 100 KBs of various sizes and query the model with questions about the KB. The results reported are the averaged values over all 500 questions.

## 6.2 EXPERIMENT RESULTS

**KBLAM attention is an accurate retriever** Given a question about the $m$th triple in the KB, we expect certain tokens in the question, particularly those containing the keyword `<name>`$_m$, to attend more strongly to the $m$th knowledge token compared to others. This would be reflected in larger values of $\tilde{w}_{n,m}^l = \langle \tilde{\boldsymbol{q}}_n^l, \tilde{\boldsymbol{k}}_m^l \rangle$ for these relevant tokens. We observe such patterns in practice, as visualized in Fig. 4. In particular, we consider a KB of 10 triples and ask KBLAM questions about a single entity, two entities, or a question without an answer in the KB. We find the post-softmax attention score at the 15th layer, i.e. the middle attention layer of the 32-layer Llama3 8B, averaged over all 32 attention heads, to correlate well with the desired entity. This property allows KBLAM to answer input questions in an interpretable way, as for any query, the top-K triples with the highest attention score form supporting evidence. This behavior suggests that KBLAM's attention mechanism functions implicitly as a retriever. We can quantitatively assess its retrieval performance by treating the attention scores, $\tilde{w}_{n,m}^l, m \in 1, \ldots, M$, as a classification score. In particular, we again extract the averaged-over-heads attention score at the 15th layer, and we evaluate the top-1 and top-5

---

[3]All KBs used for training and evaluation are released together with the paper.

accuracy, which represent the percentage of evaluation samples where the true triple received either the highest attention score or was among the top-5 highest scores, respectively. Fig. 5 presents these results, demonstrating that KBLAM's attention remains highly accurate even with a large number of triples, both for in-distribution and out-of-distribution (OOD) data. Notably, this performance is achieved *solely* through instruction tuning using paired data, without any explicit regularization or retrieval objectives. In fact, imposing regularization on the attention structure would be challenging and potentially counterproductive as it is unclear, e.g. which tokens, attention heads or attention layers should be regularized.

**KBLAM can reason about knowledge** We consider querying the model with three types of questions: Simple, two-entities (multi-entities Q&A involving two triples), and open-ended Q&A. For the first two types, we evaluate the output quality using BERT score (Zhang* et al., 2020). For open-ended questions, we use GPT-4 to score the output quality between 0 and 5 (full prompt presented in Appendix. G.3). Results are presented in Fig. 6a and Fig. 6b. Broadly, we can see that on the synthetic dataset, KBLAM shows performance comparable to in-context learning, but with a much smaller memory footprint (Fig. 3) and hence better scalability. On Enron, the out-of-distribution (OOD) data, KBLAM shows degraded performance, but still performs better than zero-shot baseline, indicating that KBLAM is using the KB information effectively. The observations from sample outputs on open-ended Q&A (Appendix. H) also align with the metrics values: On synthetic KB, KBLAM gives answer almost identical to origin triple content (Appendix. H.1), whereas on Enron, KBLAM's answer captures the meaning of the original text, with more pronounced difference in wording and details (Appendix. H.2). Note that Enron has `<value>`s

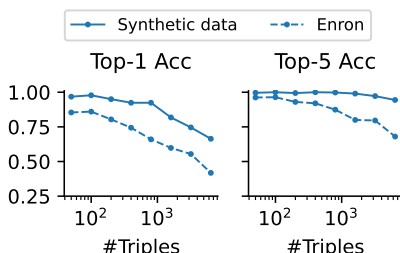

Figure 5: **Through instruction tuning, the attention shows retrieval behavior.** Given simple Q&A on the validation set of the synthetic data (solid line) and Enron dataset (OOD, dashed line), we use the attention score at the 15th layer, averaged over all attention heads, as a classification score for each triple and measure the top-1 and top-5 accuracy. KBLAM assigns the highest attention score to the truly relevant triple most of the time (with performance degraded but still reasonable on OOD).

that look entirely different from that of the synthetic data (see Table. 1 v.s. 2 in Appendix. E). We therefore believe such performance degradation could be alleviated by using a larger and more diverse synthetic dataset or through a synthetic dataset derived from a real-world dataset.

**KBLAM knows when it cannot answer a question** If the model can refuse to answer a question in case it cannot find relevant information in the context, then the risk of model hallucinations can be largely controlled. We consider a setting where, given a KB, we ask the model 100 questions in total, out of which 80 questions are answerable, and the other 20 are not. We then consider the unanswerable questions as the positive class and answerable questions as negative. To evaluate the model, we use standard binary classification metrics: precision and recall. We compare the performance of KBLAM with in-context learning, where we explicitly prompt the LLM with the following instructions: "if relevant information cannot be found in the text, please respond I am sorry I cannot find relevant information in the KB". The results are presented in Fig. 6c. Broadly, we notice that when the number of triples increases, both KBLAM and in-context learning show decreased precision, i.e. the models start to incorrectly refuse for having not found the information, however, in-context learning shows more drastic degradation. The recall for both methods stays constant, i.e. both methods do not start to hallucinate as we add more triples.

**Ablation study** We conduct ablation studies over the design choice of KBLAM and present the results in Appendix. B: We find that KBLAM also works with open-sourced sentence encoders, with exact performance depending on the encoders' capacity (Fig. 7); The frequency of adding knowledge tokens can be set less than every layer, which further improve efficiency but could cause performance drop (Fig. 8). Lastly, we study the role of knowledge tokens at each layer, where we find that knowledge tokens in earlier layers may serve as "soft prompt" for instruction following (Fig. 9).

# 7 CONCLUSION

In this paper, we propose KBLAM, which efficiently augments pre-trained LLMs with external knowledge bases. KBLAM represents external knowledge as dense continuous vectors and leverages the independence structure between triples, which offers several advantages: avoiding expensive

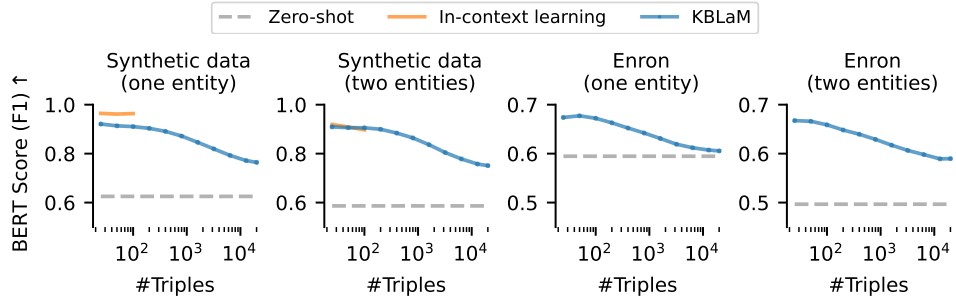

(a) **Simple and two-entity Q&A** Evaluated by BERT score (F1), which measures word-level cosine similarity between model outputs and reference answers by on embedding space. In-context learning does not have a training process, showing identical performance on Enron and synthetic data, we omitted its Enron results for graph visibility.

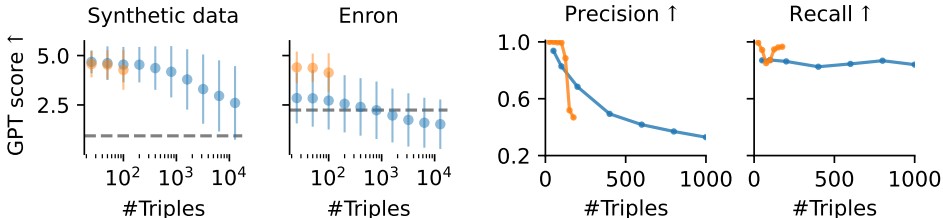

(b) **Open-ended Q&A** Scored by GPT-4 between 0 and 5, dot and bar shows the mean and standard error over 5 random seeds.

(c) **Unanswerable questions** Precision and recall for detecting questions without relevant information in the KB.

Figure 6: **KBLAM can reason about the KB.** (a) and (b): Performance on Q&A tasks evaluated by BERT and GPT-4 scores. KBLaM shows comparable quality to in-context learning on synthetic data while using less memory (Fig. 3). Some degradation occurs with Enron data, which is out-of-distribution. Zero-shot learning fails to provide sensible answers as the questions are not answerable using model's internal knowledge; (c): We consider unanswerable questions with no relevant triples in the KB as positive class (20 percent of 100 questions) and answerable questions as negative class. We then use precision and recall to measure whether the model knows to say that it "Cannot find relevant information" when it is supposed to say so. Both in-context learning and KBLAM show "over-refusal", i.e. failing to respond to answerable questions, when the number of triples grows larger (i.e. degradation in precision), however, KBLAM degrades slower.

self-attention over large knowledge sources, enabling dynamic knowledge updates without fine-tuning, improving interpretability, and allowing attention to perform retrieval. These benefits could potentially also be applied to settings using raw text as knowledge representation.

# 8   LIMITATIONS AND FUTURE WORK

**Higher quality synthetic KB** In the paper, we demonstrate the feasibility of training KBLAM with fully synthetic data. However, as shown in Fig. 6a and 6b, KBLAM generalizes to out-of-distribution dataset (Enron) but the performance degrades. We believe this can be improved through a larger and more diverse synthetic dataset. It is also possible to generate the synthetic data seeded from real KB data, such that the adapters can better capture the distribution of the real text.

**Information loss in knowledge token** KBLAM encodes sentences in the triple as a fixed-length vector, therefore KBLAM may fail to *precisely* generate the text word by word. This is acceptable for properties we considered such as "description" or "objectives". However, this is problematic for cases when we need exact names or numerical values. One future direction is to introduce a hyperparameter that adjusts the compression rate, to accommodate various amounts of information in a triple.

**More sophisticated instruction tuning** Future work could explore complex instruction tuning examples for KBLAM, such as multi-hop, multi-step, or chain-of-thought reasoning about the KB. Since KBLaM incorporates all KB information in the context, similar to in-context learning, we believe it has the potential for more flexible KB reasoning compared to RAG.

## ACKNOWLEDGMENTS

We thank Mathew Salvaris for his valuable contribution to polishing our work and assistance in cleaning up and open-sourcing the codebase.

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

# A    EXTENDED RELATED WORK

In this section, we present additional discussion on work related to KBLAM.

**Memory augmented language models**    Some other works have also incorporated external memory, i.e. a collection of fixed length latent representations, into language models (Grave et al., 2016; Dai et al., 2019; Khandelwal et al., 2020; Wu et al., 2022). KBLAM's knowledge tokens can also be seen as external memory generated from the KB. The most similar approach along this line of work is Wu et al. (2022), which constructs external memory as key-value vector pairs, similar to KBLAM. However, they require training a transformer from scratch and the memory key-value pairs come from training tokens seen in the past, whereas KBLAM's memory is from an external KB and is augmented into a pre-trained model at inference time.

**Token compression**    KBLAM's encoder transforms each triple into a continuous vector pair equivalent to 1 token in length, which essentially compresses information from the multiple-token string into one single token. There have been some recent works aiming at compressing a long context into more continuous representations (Li et al., 2024b). Ge et al. (2024) replaces short prompts with special tokens to instruct the LLM to do some particular tasks, however they require fine-tuning of the LLM. Mu et al. (2024) encode a long context into memory slots through a fine-tuned LLM, which is a more powerful but also costly compressor, which we believe could in future work be combined with KBLAM's framework as an alternative to pre-trained sentence encoder.

**Multi-modal language model**    Recent advances in Multi-modal Language Models (Liu et al., 2023; Gao et al., 2023; Liu et al., 2024a; Zhao et al., 2023; Wu et al., 2023, MLMs), demonstrate the possibility of incorporating data of other modalities (e.g. images, videos) into a pre-trained LLM through learned adapters and instruction tuning. With KBLAM, we can consider a KB as another modality in that a KB is presented to the LLM in the form of continuous vectors via an encoder, and an encoder-adapter is learned also through instruction tuning inspired by MLM literature.

**Augmenting information into an LLM in a continuous form**    RAG and in-context learning incorporates external information into LLMs in the form of a *discrete string*, whereas KBLAM augments information into LLMs as fixed length *continuous* representations. Many works also consider first encoding external documents as continuous vectors, then concatenating the vectors as the input for the decoder model (Izacard & Grave, 2020; Izacard et al., 2023; Ye et al., 2023) or using cross-attention to inject these vectors into the decoder (Borgeaud et al., 2022). However, these works focus on training a new model from scratch, while KBLAM is applied to augment a *pre-trained* LLM with knowledge. Concurrent works (Li et al., 2025; Zhuang et al., 2025) consider augmenting few-shot demonstration information into a pre-trained LLM in the form of continuous vectors, however, KBLAM focuses on augmenting factual knowledge from a large scale KB rather than task-solving ability via few-shot examples. Recent work (Yen et al., 2024) also considers augmenting a *pre-trained* LLM with encoded information, similar to our setting. However, their approach requires training an encoder *from scratch*, whereas KBLAM's only trains linear adapters after a pre-trained black-box encoder. In addition, they add extra cross-attention layers after each self-attention layer for augmentation, while KBLAM modifies existing self-attention layers without introducing new components.

# B    ABLATION STUDY

We conducted a number of ablation studies on the design choices of KBLAM.

**Choice of encoders**    KBLAM relies on a pre-trained sentence encoder, and the capacity of the encoder can affect the performance. Indeed our results (Fig. 7) verify this: We experimented with some open-sourced alternatives from sentence transformer (Reimers & Gurevych, 2019). Broadly we observe that the commercial OpenAI embedding works better than the open-sourced one, models with higher embedding dimensions perform better than those with lower dimensions.

**Where to add knowledge tokens**    In Fig. 4 and 5, we noted that the attention score from the 15th layer exhibits accurate retrieval behavior. Extending this analysis to other layers (Fig. 9), we find that they *do not* demonstrate such straightforward accuracy (although further probing might reveal more nuanced patterns). We suspect that this indicates that the amount of KB information provided by the knowledge tokens vary at different layers provide. To better understand this, we examining

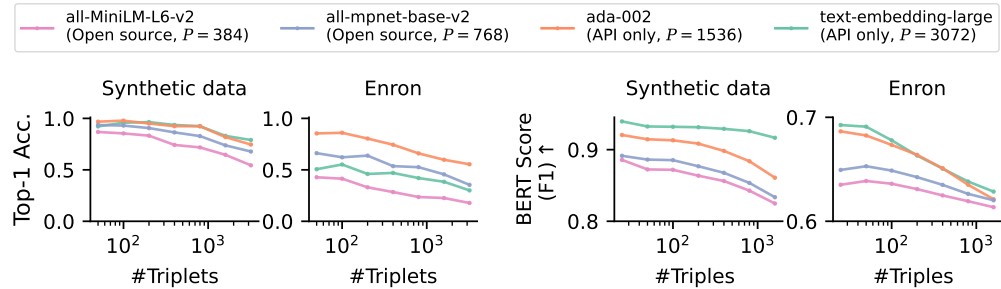

Figure 7: **KBLAM works with other types of pre-trained encoders.** Similar to the setting in Fig. 5 and Fig. 6a, but with sentence encoder of varying capacity.

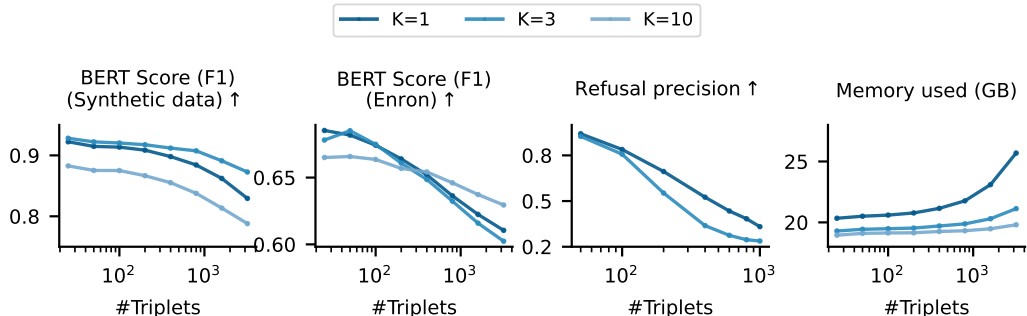

Figure 8: **Ablation study on knowledge token injection frequency (K).** If we have knowledge tokens added to attention , the model performance drops ($K = 10$), but the memory overhead decreases and efficiency increases. The BERT score of $K = 3$ is comparable to $K = 1$ (first two columns), however when $K = 3$, the model shows lower precision at refusing answering unanswerable questions (third column), and when $K = 10$, the model fails to output refusal answers.

the variation of keys and values at each layer ($\mathbb{V}_m[\tilde{\boldsymbol{k}}_m^l]$ and $\mathbb{V}_m[\tilde{\boldsymbol{v}}_m^l]$) across triples, we observe that encoder outputs vary minimally in earlier layers.

**Frequency of knowledge tokens**   The previous observation indicates that the knowledge tokens may not be providing information about the KB at every layer, therefore we consider varying the frequency of adding knowledge tokens (denoted by $K$). That is, for certain attention layers of KBLAM, we used Eq. (3) instead of Eq. (9). In particular, we considered only adding knowledge every 1, 3 and 10 layers and the results are presented in Fig. 8. Broadly, we notice with lower frequency, the model tends not to follow instructions: When $K = 10$, KBLAM fails to provide any refusal answer. As such we suspect that knowledge tokens in earlier layers may serve as soft instruction prompts that guides the LLM how to use the KB.

## C   EXTENDED EXPERIMENT RESULTS

**Comparison between KBLAM's retrieval performance and BM25**   We also ran experiments with BM25, lexical search algorithm, as a baseline for the top-1 and top-5 retrieval accuracy, and the results can be seen in Figure 10a. We find that, on the synthetic dataset, KBLaM achieves comparable top-5 accuracy to BM25 across all KB sizes.

As with KBLAM, BM25's performance also degrades on Enron, although the degradation is less sever, leading to a larger gap between the performance of the two methods as the number of triples increases.

**Perturbing the synthetic dataset**   We also experiment with a setting where we *perturb* the synthetic dataset, so that the entity's name in the question no longer exactly matches the entity's name in the knowledge base (e.g. an entity with name "Euler's Edifice" in the knowledge base may be referred to

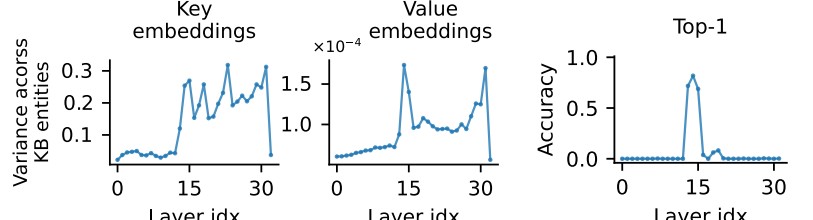

Figure 9: **Knowledge tokens' embeddings contain various level of KB information across layers.** The embeddings in earlier layers vary less among different entities, indicating that they may serve as instruction prompt on how to use the KB. Later layers show a higher degree of variation, indicating that they may be used to provide actual knowledge.

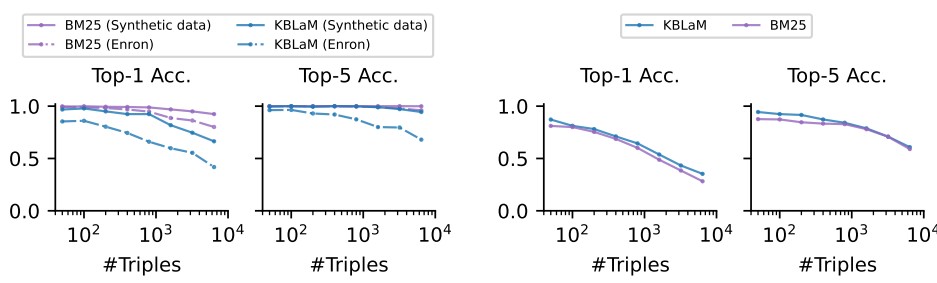

(a) Retrieval accuracy on synthetic KB and Enron.  (b) Retrieval accuracy perturbed synthetic KB.

Figure 10: **KBLaM shows retrieval accuracy comparable with BM25.** We consider a setting similar to that of Fig. 5, additionally including BM25 as baseline. On synthetic data, KBLAM's top-1 accuracy is slightly worse than BM25 but shows very close top-5 accuracy (left subfigure, solid purple and blue lines). In the perturbed synthetic dataset, where the keyword in the questions no longer matches that in the triple exactly, KBLAM shows superior performance to BM25 at all numbers of triples tested.

as "Euler's Building" in the question). This was done by feeding the entity names through GPT-4 and asking it to perturb the question as described above.

Using the same KBLAM model trained on synthetic data where there is always an exact match between the question and knowledge base, we find that despite never having seen such queries during training, KBLAM can still retrieve correctly from the knowledge base, as shown in Figure 10b. Compared to the baseline BM25, KBLAM degrades less and, unlike with the other datasets, outperforms BM25 at all KB sizes.

**Latency comparison with RAG**  In Fig. 11, we compare the RAG's time and memory overhead with KBLAM, where we focus on the overhead from the LLM assuming RAG uses a highly-optimized retriever with negligible overhead. We find that RAG, with 5 triples retrieved into the context, shows time latency and memory higher than KBLAM, due to the large number of characters in the context. KBLAM on the other hand shows stable latency with respect to the KB size, in that KBLAM's overhead from QKV projection and FFN does change with KB size.

## D CURRENT LIMITATIONS AND EXTENDED FUTURE WORK

**One-time training costs of KBLAM**  A limitation of KBLAM lies in its non-zero one-time costs (around 24-48 hours on a single 80GB GPU) compared to some other methods for knowledge augmentation such as in-context-learning, RAG, or prompt caching. Namely, KBLAM requires the fine-tuning of adapters, and the creation of triples if the input is not already organized in this format. However, the former is a one-time cost, with adapters generalizing across corpora, as seen with the Enron dataset, and the latter only needs to be performed once per corpus. These one-time costs are in contrast to the use of fine-tuning for knowledge augmentation (Hu et al., 2021; Liu et al.,

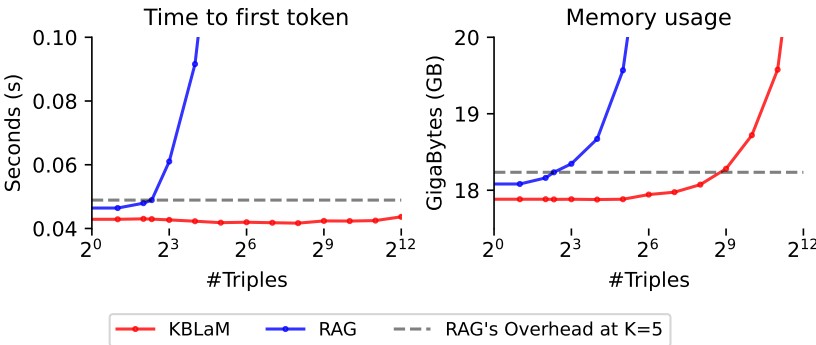

Figure 11: **KBLAM maintains lower latency and memory usage than RAG while keeping the full KB in context.** Averaged over 40 random queries from the synthetic KB, for a KB with 512 triples, KBLAM matches RAG's memory footprint which retrieves 5 triples (gray dashed line, right), while providing consistently faster token generation across all KB sizes (red line, left).

2024c) which have much higher one-time costs, as models need to be fine-tuned for each corpus. In addition, as a result of these one-time costs, the runtime costs of KBLAM are significantly lower than in-context learning, RAG, or prompt caching, as shown in Figure 11.

**Utilizing more extensive structural information in the KB.** One of the motivations behind KBLAM's design is that a KB can be viewed as a collection of key-value pairs, similar to a dictionary, therefore we can independently encode each triple and augment them into a pre-trained LLM efficiently. Future work can study designing customized attention structures for more sophisticated structures underlying a KB, e.g. tree structure, if we consider <name> as a root node and triples under same <name> but different <property>s as the leaf nodes, or graph structure, if we assume each triple as a node and edges connect two triples with <value> equal to the <name> of another triple.

# E   SAMPLE KB

Here we present some example triples from both KBs, where each triple is of format

(<name>;<property>;<value>)

We present ten examples from the synthetic KB (Table. 1) and from Enron (Table. 2) respectively.

| <name> | <property> | <value> |
|---|---|---|
| Posh Poodle | description | A line of organic teas with health-boosting properties |
| Maserati Vasco da Gama | description | A smart lighting system with customizable settings |
| ThornTactician | description | A historical strategy game set in ancient Rome. Conquer the empire |
| Bohr's Bookshelf | purpose | To promote a sense of calm and improve quality of life |
| Pixelated Prose | purpose | To raise awareness about the ethical implications of technology |
| Matrix Monument | description | A historical library housing rare manuscripts, books, and documents |
| Nova Citadel | description | A secure data center offering cloud storage and cybersecurity services |
| Celestial Genome | objectives | Test radiation shields, monitor devices, and come up with fixes |
| The Mamas and the Papas | objectives | To provide guidance, build a supportive network, and promote positive parenting practices |
| The Doom That Came to Sarnath | description | A high-stakes fashion competition featuring talented designers from around the world. It's a showcase of creativity and style |

Table 1: Synthetic KB

| <name> | <property> | <value> |
|---|---|---|
| Fill Order | description | Feature that allows traders to automatically fill any order that is at the top of the stack. |
| Allen & Overy | description | A solicitors' partnership |
| EAP PA | description | Part of the PA's list |
| Analyst Recommendations Screener | purpose | Screen for attractive stocks using Wall Street analysts' opinions |
| Financial Accounting Standard 133 | description | A standard that deals with accounting rules for derivatives and other financial instruments. |
| Vividence | description | A customer management applications firm. |
| LD Calculation Methodology | description | A methodology proposed by CDWR for consistency among contracts, viewed as more 'objective' by CDWR. |
| Boxed Trio of Rosewood Stoppers | description | Uniquely shaped rosewood stoppers protect opened wines from rapid oxidation. |
| Learning Company | description | A troubled software company sold by Mattel to a buy-out firm in exchange for a share of future profits. |
| Code 238 | description | Reflects dollars paid to a generator for DAM Contract Balancing that is Out-of-Merit for LRR. |
| Validata | purpose | Pre-employment screening services |

Table 2: Enron KB

# F   SAMPLE Q&A

An example of four types of questions used for instruction tuning is presented in Fig. 12.

**Example KB of four triples**

```
1. <Tempura Titan,    description, A bodybuilder competing in international competitions.>
2. <Pythia's Plates,  objective,   To promote holistic health and relaxation.>
3. <Samosa Sage,      purpose,     To provoke thought and discussion.>
4. <Samosa Sage,      description, A philosopher pondering the meaning of life.>
```

**Simple Q&A**

**Q**: What's the description of Samosa Sage?
**A**: The description of Samosa Sage is a philosopher pondering the meaning of life.

**Multi-entities Q&A**

**Q**: Tell me the purpose of Samosa Sage and the description of Tempura Titan.
**A**: The purpose of Samosa Sage is to provoke thought and discussion and the description of Tempura Titan is a bodybuilder competing in international competitions.

**Open-ended Q&A**

**Q**: What is the objective of Pythia's Plates and what do you think of its social impacts?
**A**: The objective of Pythia's Plates is to promote holistic health and relaxation, which could potentially lead to improved physical and mental wellbeing in their community, increased health awareness, and stronger social connections.

**Unanswerable Q&A**

**Q**: Can you tell me the description of Pesto Paladin?
**A**: Sorry I cannot find relevant information in the KB.

Figure 12: **Examples of instructions tuning dataset.** Given a KB of 4 triples (top panel), we generate 4 types of Q&A for instruction tuning. The KB shown is sampled from the synthetic KB; The question and answer for open-ended Q&A are also synthesized by GPT, the rest Q&As are generated using formatted string.

# G PROMPT

In this section, we provide all prompts used for KBLAM's experiments.

## G.1 PROMPT FOR SYNTHETIC KB GENERATION

To prompt the GPT-4 for generating a synthetic KB, we begin by setting a system prompt as

```
You are a AI system that generates synthetic data examples in JSON format
```

Then we construct a list of idea types and a list of object types:

```
idea_types = [
    'greek letters', 'fiction characters', 'famous rock bands',
    'birds', 'animals', 'natural phenomena', 'physical locations',
    'artist names', 'classical music', 'musical instruments',
    'music genres', 'art styles', 'ancient Roman concepts',
    'Hindu myths', 'Cthulhu Mythos', 'real-world company names',
    'mythological creatures', 'planets and stars', 'historical figures',
    'literary genres', 'botanical names', 'famous landmarks',
    'scientific concepts', 'space missions', 'inventions',
    'philosophical terms', 'chemical elements', 'famous scientists',
    'marine life', 'mythological places'
]

object_types = [
    'education company', 'tech company', 'car company',
    'entertainment company', 'construction company', 'retail company',
    'finance company', 'healthcare company', 'restaurant', 'hotel',
    'github repo', 'project', 'meeting room', 'building', 'lab',
    'airline', 'textbook', 'website', 'personal blog',
    'gaming company', 'consulting firm', 'biotech company', 'app',
    'software tool', 'bookstore', 'e-commerce site',
    'social media platform', 'fitness brand', 'fashion brand',
    'non-profit organization'
]
```

Then for each combination of object type and idea type in the list, we prompt GPT with

```
Please randomly generate 50 {object_type} name innovated by {idea_type}.
The name should have various styles.
The generated name should be of diverse style and length, for example the
 name could have hyphen, space in it or multiple words.
```

which gives us 50 different names. Then in the same context, we further prompt GPT with

```
Now for each of the names generated, generate a short desciption, short
objectives, and a purpose for the data.
Please ensure that the generated contents has **LOW** correlation with
the name.
```

which generates the `<value>` for three `<property>`s: description, objectives and purpose. Lastly, we perform one round of polishing, to let GPT diversity the language and style of the generated KB triple

```
Now for each of the name, description, objective and purpose generated,
make their text style more diverse using a mixture of formal and informal
 language.
```

## G.2   PROMPT FOR OPEN-ENDED Q&A GENERATION

To generate open-ended Q&A instructions, we first take a triple from the KB, construct a simple Q&A from it, and then we feed the simple Q&A into GPT and let GPT augment it into a more complex form via the following prompt

```
You are given a question and answer pair, please extend the question to
be open-ended and generate a short answer. For example, you could
generate
What is the objective of xxx and what do you think of it?
Make sure the answer is **only** based on information provided from the
QA pair. In addition, please generate in the format of:
Q: ...
A: ...
```

## G.3   PROMPT FOR GPT EVALUATION OF OPEN-ENDED Q&A

To evaluate the answer quality of open-ended Q&A tasks, we again use GPT. In particular, we ask GPT to score the answer from two aspects: 1. How grounded the answer is given the question and the KB; 2. How reasonable the open-ended part of the answer is.

We begin with a system prompt of

```
You are an AI system that evaluates the quality of generated responses.
Your goal is to return a score between 0 and 5 indicating how accurate
and useful the response is. An accurate and useful response should get a
high score of 5.
```

Then, we use the following prompt, which encourages the model to give chain-of-thought reasoning for scoring (Zheng et al., 2023):

```
A model is given a question about some information and evidence. The
question is composed of two parts, a part that involves repeating
information in the evidence and a part that potentially involves open-
ended thinking. Then the model generates a response. Evaluate the
response based on how grounded it is given the evidence and how
reasonable it is. Return an integer score and step by step explanation of
 how you arrived at the score. Score of 5 means the response is accurate,
 relevant and reasonable (in that it meets common sense).
```

```
If the response addresses the question and uses the evidence in a
relevant way, it should get a high score of 5. Score of 0 means the
response is inaccurate and irrelevant or model is hallucinating. Score
between 0 and 5 means the response is partially correct and relevant.
```

Followed by the prompt, we include 5 examples to better help GPT calibrate the scoring rule

```
Example 1:
Evidence: "The purpose of Alexandria is to extract knowledge."
Question: "Describe the purpose of Alexandria and how it can benefit
users."
Model output: "The purpose of Alexandria is to extract knowledge, it can
benefit users by providing a structured way to organize knowledge."
Score: 5
Reason: The model's response is accurate and relevant to the question and
 evidence, the open-ended part is reasonable.

Example 2:
Evidence: "The purpose of Alexandria is to extract knowledge."
Question: "Describe the purpose of Alexandria and what can it extract."
Model output: "The purpose of Alexandria is to extract knowledge, it can
extract knowledge knowledge."
Score: 5
Reason: The model's response is accurate and relevant to the question and
 evidence.

Example 3:
Evidence: "GreatTool is an app that helps users to be more productive."
Question: "Describe GreatTool and how it may affect the community."
Model output: "GreatTool is an app that helps users to be more productive
. It may affect the community by helping users to sleep better."
Score: 3
Reason: The model's response is accurate and relevant to the question and
 evidence but it is not very reasonable.

Example 4:
Evidence: "GreatTool is an app that helps users to be more productive."
Question: "Describe GreatTool and how it may affect the community."
Model output: "GreatTool is an app that helps users to be more productive
. It may affect the community by helping users to organize their tasks
and manage their time better improving their productivity."
Score: 5
Reason: The model's response is accurate and relevant to the question and
 evidence and the open ended part is sensible and reasonable.

Example 5:
Evidence: "GreatTool is an app that helps users to be more productive."
Question: "Describe GreatTool and how it may affect the community."
Model output: "GreatTool is great tool with many feature"
Score: 0
Reason: The model's response is not accurate and doesn't answer the
question.
```

### G.4 PROMPT FOR LLAMA EVALUATION

For in-context learning, we use the following prompts for evaluation

```
# Simple Q&A
Please answer questions based on the given text with format: "The {
property} of {name} is {description}"

# Two-entity Q&A
```

```
Please answer questions based on the given text with format: "The {
property} of {name1} is {description}; The {property} of {name2} is {
description}; ..."

# Open-ended Q&A
You are provided a context and a question that has a retrieval part and
an open-ended part.
Please answer the question based on the given text.
If the information for the open-ended part is not provided in the context
, please generate a potential possible answer.

# Unanserable questions
Please answer questions based on the given text with format: "The {
property} of {name} is {description}", if relevant information cannot be
found in the text, please respond "I am sorry I cannot find relevant
information in the KB".
```

For zero-shot learning, we use the following prompt

```
# Simple Q&A
Please answer the question in a very compact manner with format: The {
property} of {name} is {description}

# Two-entity Q&A
Please answer the question in a very compact manner with format: "The {
property} of {name1} is {description}; The {property} of {name2} is {
description}; ...

# Open-ended Q&A
Please answer the question based on your knowledge.
```

### G.5    QUESTION TEMPLATE

For simple Q&A, we use the following templates

```
What <property> does <name> have?,
What is the <property> of <name>?,
Tell me about the <property> of <name>.,
Can you let me know the <property> of <name>?,
Can you inform me about the <property> of <name>?,
Describe the <property> of <name>.,
What details can you share about the <property> of <name>?,
What kind of <property> does <name> have?,
Provide details on the <property> of <name>.,
What features does the <property> of <name> include?,
Can you elaborate on the <property> of <name>?,
How would you describe the <property> of <name>?,
What can you tell me about the <property> characteristics of <name>?,
Can you explain the <property> of <name>?,
What insights can you provide about the <property> of <name>?,
What should I know about the <property> of <name>?,
```

For multi-entities Q&A involving $G$ triples, $\{(\text{<name>}_g, \text{<property>}_g, \text{<value>}_g)\}_{g=1}^{G}$, we use the following template

```
What is the {}
Tell me {},
Can you let me know {},
Can you inform me {},
Describe {},
Explain {},
Could you describe the {},
What can you tell me about {},
```

```
Could you provide information on {},
Please enlighten me about {},
Can you clarify {} for me?,
Could you give me a detailed description of {},
I need more information on {},
```

then for each blank in the curly brackets, we fill in the string

```
the <property>_1 of <name>_1, ..., the <property>_g of <name>_g, ... and
the <property>_G of <name>_G
```

## H  SAMPLE OUTPUT

In this section, we present sample output from the model. In particular, we present generated answers from KBLAM, in-context learning, and zero-shot learning, on open-ended Q&A tasks for synthetic KB (Appendix. H.1) and Enron (Appendix. H.2), given questions about a "Relevant triple" from a KB of 100 triples.

Broadly, we can see that zero-shot consistently produces hallucinations, as the contents of both the synthetic KB and Enron KB differ significantly from the real-world factual knowledge stored in the LLM's parameters.

KBLAM and in-context learning, on the other hand, utilize the information from the KB to synthesize the answer. In-context learning provides highly precise answers to all questions, however it incurs substantial computational overhead and struggles to scale to larger KBs On synthetic data, KBLAM accurately captures the information from the KB in terms of meaning and sentence structure, though it may use slightly different wording, e.g. in the last example on "Titdal surge", KBLAM uses "renewable" instead of "sustainable" as in the original triple.

When tested on the out-of-distribution Enron data, KBLAM still grasps the general meaning of the relevant triple, albeit with more noticeable differences. As discussed in the limitations section of the main text, we believe this performance gap can be narrowed by using a more diverse synthetic KB with a stronger real-world foundation during training.

### H.1  SYNTHETIC KB

```
Relevant triple:
(ReefPulse Jellyfish; objectives; to provide real-time alerts, support
remote monitoring, and improve home safety)

Question:
Can you inform me about the objectives of ReefPulse Jellyfish and what do
 you think of its effectiveness in achieving these goals?

KBLaM output:
The objectives of ReefPulse Jellyfish are to provide real-time monitoring
, send alerts, and improve home safety. While the effectiveness of
ReefPulse Jellyfish in achieving these goals would depend on various
factors such as the accuracy of the monitoring technology, the
reliability of the alert system, and the user experience, the outlined
objectives suggest a comprehensive approach to enhancing home safety.

In-context learning output:
According to the context, the objectives of ReefPulse Jellyfish are to
provide real-time alerts, support remote monitoring, and improve home
safety.
Based on these objectives, I think ReefPulse Jellyfish is effective in
achieving its goals. The real-time alerts feature allows users to stay
informed and take prompt action in case of any issues, which can help
prevent potential hazards and ensure a safer home environment. The remote
 monitoring feature enables users to keep track of their home's
```

```
conditions from anywhere, which can be particularly useful for people who
 are away from home for extended periods. Additionally, the improved home
 safety feature can help reduce the risk of accidents and injuries,
making it a valuable tool for homeowners.
Overall, I believe ReefPulse Jellyfish is
```

**Zero-shot learning output:**
```
ReefPulse Jellyfish is a product designed to monitor and control the
water parameters in aquariums, specifically targeting jellyfish and other
 delicate marine life. Based on my knowledge, the objectives of ReefPulse
 Jellyfish are:

1. **Accurate Water Parameter Monitoring**: To provide real-time
monitoring of water parameters such as pH, ammonia, nitrite, nitrate, and
 temperature, ensuring that the aquarium's water conditions are optimal
for the jellyfish and other inhabitants.
2. **Automated Water Parameter Control**: To automatically adjust water
parameters to maintain a stable and healthy environment, reducing the
need for manual intervention and minimizing the risk of water parameter
fluctuations.
3. **Enhanced Jellyfish Care**: To provide a more precise and controlled
environment
```

---

**Relevant triple:**
```
(Sagan's Starry Hotel; objectives; to create a magical and enchanting
environment for guests)
```

**Question:**
```
What features does the objectives of Sagan's Starry Hotel include and how
 do these features contribute to the overall guest experience?
```

**KBLaM output:**
```
The objectives of Sagan's Starry Hotel is to create a magical and
enchanting environment for guests. These features likely include themed
decor, special amenities, and exceptional service, all designed to
transport guests into a fantastical world, enhancing their stay and
making it memorable.
```

**In-context learning output:**
```
Based on the context, the objectives of Sagan's Starry Hotel are to
create a magical and enchanting environment for guests. The features that
 contribute to this objective include:
* Creating a magical and enchanting environment for guests
* Providing a unique and memorable experience for guests
* Enhancing the overall guest experience through attention to detail and
a focus on customer satisfaction
These features contribute to the overall guest experience by creating a
sense of wonder and excitement, making guests feel special and valued,
and providing a memorable and enjoyable stay.
```

**Zero-shot learning output:**
```
I apologize, but I couldn't find any information on "Sagan's Starry Hotel
". Carl Sagan was an American astronomer and science communicator, but I
couldn't find any evidence of a hotel bearing his name.

If you meant to ask about a different hotel or a hypothetical hotel
concept, please provide more context or details, and I'll do my best to
answer your question.
```

---

**Relevant triple:**
```
(Currents Coach; purpose; to help you stay active and achieve your
fitness goals)
```

**Question:**

```
Describe the purpose of Currents Coach and how it can benefit users in
their daily lives.
```

**KBLaM output:**
```
The purpose of Currents Coach is to help individuals track and improve
their fitness. By using Currents Coach, users can monitor their physical
activities, set fitness goals, and track their progress over time. This
can lead to a more structured and effective fitness routine, ultimately
contributing to better overall health and well-being.
```

**In-context learning output:**
```
The purpose of Currents Coach is to help individuals track and improve
their fitness. This app can benefit users in their daily lives by
providing a personalized and comprehensive fitness tracking system. With
Currents Coach, users can set and achieve their fitness goals, monitor
their progress, and receive guidance and support to help them stay
motivated and on track.

Here are some potential benefits of using Currents Coach:

* Improved physical health: By tracking and monitoring their fitness
progress, users can make data-driven decisions to improve their overall
physical health and well-being.
* Increased motivation: Currents Coach provides a sense of accountability
 and motivation, helping users stay committed to their fitness goals and
make progress towards achieving them.
* Personalized guidance: The app offers personalized guidance and support
```

**Zero-shot learning output:**
```
Currents Coach is a personalized financial wellness platform designed to
help users manage their finances, achieve their financial goals, and
improve their overall financial well-being. The purpose of Currents Coach
 is to provide users with a comprehensive and interactive financial
planning experience that is tailored to their individual needs and goals.

Here are some ways Currents Coach can benefit users in their daily lives:

1. **Personalized Financial Planning**: Currents Coach uses machine
learning algorithms to analyze users' financial data and provide
personalized financial plans, including budgeting, saving, and investing
recommendations.
2. **Goal Setting and Tracking**: Users can set and track their financial
 goals, such as saving for a down payment on a house, paying off debt, or
 building an emergency fund. Current
```

**Relevant triple:**
```
(Hertz Horizons; description; a music production company specializing in
independent artists)
```

**Question:**
```
Can you explain the description of Hertz Horizons and what kind of
artists they specialize in?
```

**KBLaM output:**
```
The description of Hertz Horizons is a music production company
specializing in independent artists. They focus on supporting and
promoting independent musicians, providing them with the resources and
opportunities needed to succeed in the music industry.
```

**In-context learning output:**
```
According to the context, Hertz Horizons is a music production company
specializing in independent artists.
```

**Zero-shot learning output:**

```
Hertz Horizons is a music production and distribution company that
specializes in creating and releasing music for film, television, and
other visual media. They are known for their unique approach to music
production, which involves creating custom music tracks for specific
scenes or moments in a film or TV show.

Hertz Horizons is particularly known for their work in the electronic and
 ambient genres, and they have worked with a wide range of artists and
producers to create music for various projects. Some of the artists they
have worked with include:

* Tycho: A electronic music producer and composer who has worked on
numerous film and TV projects, including the Netflix series "Stranger
Things".
* Rival Consoles: A British electronic music producer and composer who
has worked on
```

**Relevant triple:**
(Tidal Surge; objectives; to generate clean and sustainable energy)

**Question:**
Provide details on the objectives of Tidal Surge and explain how it aims
to achieve them.

**KBLaM output:**
The objectives of Tidal Surge is to generate clean and renewable energy.
It aims to achieve these objectives by harnessing the power of tidal
currents, which are a consistent and powerful source of energy. This
involves the development of advanced technologies and infrastructure
designed to capture and convert the kinetic energy of the tides into
electricity.

**In-context learning output:**
According to the context, the objectives of Tidal Surge are to "generate
clean and sustainable energy".

To achieve these objectives, Tidal Surge likely aims to harness the power
 of tidal currents, which are the regular rise and fall of the sea level
caused by the gravitational pull of the moon and sun. This renewable
energy source has the potential to provide a significant portion of the
world's energy needs while reducing greenhouse gas emissions and
mitigating climate change.

To generate clean and sustainable energy, Tidal Surge may employ various
technologies, such as:

1. Tidal barrages: These are structures that harness the energy of tidal
currents by creating a dam-like structure across a tidal estuary or a
narrow channel.
2. Tidal stream generators: These are

**Zero-shot learning output:**
Tidal Surge is a research project focused on developing a novel tidal
energy converter (TEC) that can harness the power of ocean tides to
generate electricity. The primary objectives of Tidal Surge are:

1. **Design and development of a more efficient TEC**: The project aims
to design and develop a TEC that can capture a larger portion of the
tidal energy available in the ocean, thereby increasing the overall
efficiency of the device.
2. **Improved scalability and cost-effectiveness**: Tidal Surge aims to
create a TEC that can be easily scaled up or down depending on the
specific tidal energy resource and the needs of the power grid. This will

> help reduce the cost of tidal energy generation and make it more
> competitive with other renewable energy

## H.2 ENRON

**Relevant triple:**
(Sitraka; description; a provider of Internet software technologies for
IT organizations)

**Question:**
What kind of description does Sitraka have and what services do they
offer?

**KBLaM output:**
The description of Sitraka is a software company providing networking
solutions for businesses. They offer services that likely include
developing and implementing network infrastructure, managing network
security, and possibly providing technical support to ensure smooth
network operations for their clients.

**In-context learning output:**
Based on the given text, Sitraka is described as a provider of Internet
software technologies for IT organizations.

As for the services they offer, the text does not provide specific
information. However, based on the general description, it can be
inferred that Sitraka offers software technologies and solutions for IT
organizations, possibly related to internet-based services, network
management, or other IT-related areas.

**Zero-shot learning output:**
According to my knowledge, Sitraka is a software company that provides IT
 service management solutions. They offer a range of services and
products that help organizations manage and optimize their IT
infrastructure, including:

* IT Service Management (ITSM) software: Sitraka's flagship product is
the ITSM software, which provides a comprehensive platform for managing
IT services, including incident, problem, and change management.
* Asset Management: Sitraka's asset management solution helps
organizations track and manage their IT assets, including hardware,
software, and network devices.
* Service Desk: Sitraka's service desk solution provides a centralized
platform for managing IT requests, incidents, and problems.
* Reporting and Analytics: Sitraka's reporting and analytics solution
provides real

**Relevant triple:**
(Unocal; objectives; secure OPIC funds for offshore oil and natural gas
projects)

**Question:**
What features does the objectives of Unocal include and how do these
features impact their operations?

**KBLaM output:**
The objectives of Unocal include securing offshore oil and gas reserves.
These features impact their operations by ensuring that they can access
and extract these natural resources efficiently and safely, which is
crucial for meeting energy demands and maintaining a stable supply chain.

**In-context learning output:**
Based on the provided context, the objectives of Unocal are:

```
* Secure OPIC funds for offshore oil and natural gas projects

These objectives do not provide specific features, but rather outline the
 company's goals and priorities. However, we can infer that securing
funds for offshore oil and natural gas projects may involve features such
 as:

* Financial planning and budgeting
* Risk assessment and management
* Project development and execution
* Regulatory compliance and permitting
* Collaboration with stakeholders, including government agencies and
investors

These features would likely impact Unocal's operations by:

* Guiding the allocation of resources and budgeting for specific projects
* Informing the company's strategic decisions and priorities
* Influencing the company's relationships with stakeholders and partners

Zero-shot learning output:
Unocal, a multinational energy corporation, has several objectives that
shape its operations. Based on my knowledge, the following features are
included in Unocal's objectives and their impact on operations:

1. **Safety and Environmental Protection**: Unocal prioritizes the safety
 of its employees, contractors, and the public, as well as environmental
protection. This objective is reflected in their commitment to reducing
environmental impacts, implementing safety protocols, and conducting
regular training programs. Impact: This objective ensures a safe working
environment, reduces the risk of accidents, and minimizes environmental
damage.
2. **Operational Excellence**: Unocal aims to optimize its operations by
improving efficiency, reducing costs, and increasing productivity. This
objective is achieved through process improvements, technology adoption,
and employee
```

```
Relevant triple:
(San Francisco Symphony; purpose; concert organization)

Question:
Provide details on the purpose of San Francisco Symphony and what
activities they engage in to fulfill this purpose.

KBLaM output:
The purpose of San Francisco Symphony is to organize community concerts.
To fulfill this purpose, they likely engage in activities such as
planning and coordinating concert events, recruiting musicians, promoting
 the events to the community, and possibly collaborating with local
organizations to ensure the concerts are accessible and enjoyable for a
wide audience.

In-context learning output:
Based on the given text, the purpose of San Francisco Symphony is a
concert organization. This implies that the San Francisco Symphony is a
music organization that aims to promote and perform classical music,
likely through concerts and other musical events.

To fulfill this purpose, the San Francisco Symphony may engage in various
 activities such as:

* Organizing and performing concerts, including classical music
performances, symphony orchestras, and chamber music recitals
```

```
* Producing and presenting musical events, such as opera, ballet, and
dance performances
* Providing music education and outreach programs, such as music classes,
 workshops, and community outreach initiatives
* Collaborating with other music organizations, schools, and community
groups to promote music education and appreciation
* Developing and maintaining a professional

Zero-shot learning output:
The San Francisco Symphony (SFS) is a world-renowned orchestra based in
San Francisco, California. The purpose of the San Francisco Symphony is
to provide high-quality musical performances, promote music education,
and foster a sense of community through music.

To fulfill this purpose, the San Francisco Symphony engages in a variety
of activities, including:

1. Concerts: The SFS performs over 200 concerts each year, featuring a
wide range of repertoire from classical masterpieces to contemporary
works. These concerts take place at Davies Symphony Hall, the orchestra's
 home venue, as well as at other locations throughout the Bay Area.
2. Music Education: The SFS offers a range of music education programs
for children and adults, including the SFS Youth Orchestra
```

```
Relevant triple:
(TBWA/Chiat/Day; objectives; eliminate private office and individual
cubicle,Promote collaboration)

Question:
What insights can you provide about the objectives of TBWA/Chiat/Day and
what do you think of their approach to promoting collaboration?

KBLaM output:
The objectives of TBWA/Chiat/Day is promote face-to-face collaboration
and eliminate conference rooms. Their approach to promoting collaboration
 by eliminating conference rooms suggests a focus on more informal and
spontaneous interactions, which can be beneficial for fostering
creativity and open communication among team members.

In-context learning output:
Based on the provided context, the objectives of TBWA/Chiat/Day are:

* Eliminate private office and individual cubicle
* Promote collaboration

Insights that can be provided about the objectives of TBWA/Chiat/Day are:

* The agency is focusing on creating a more collaborative work
environment by eliminating private offices and individual cubicles. This
suggests that they value teamwork and communication among employees.
* By promoting collaboration, TBWA/Chiat/Day is likely trying to increase
 creativity, innovation, and productivity among its employees. This
approach can lead to better ideas, faster problem-solving, and improved
morale.

As for their approach to promoting collaboration, it seems that they are
taking a bold step by eliminating private offices and

Zero-shot learning output:
TBWA/Chiat/Day is a global advertising agency that was founded in 1978 by
 Jay Chiat and Richard Laermer. The agency is known for its creative and
innovative approach to advertising, and has worked with a wide range of
clients across various industries.
```

As for their objectives, TBWA/Chiat/Day's mission is to "Challenge the Status Quo" and to "Change the Game" for their clients. They aim to create advertising that is not only creative and effective, but also memorable and impactful. The agency's approach is centered around the idea of "Disruption," which involves challenging the conventional norms and expectations of advertising and finding new and innovative ways to connect with consumers.

In terms of promoting collaboration,

---

**Relevant triple:**
(Go2net Mail; purpose; used for sending and receiving emails)

**Question:**
Describe the purpose of Go2net Mail and explain how it benefits users.

**KBLaM output:**
The purpose of Go2net Mail is used for sending and receiving emails. It benefits users by providing a reliable and efficient platform for communication, ensuring that emails are delivered promptly and securely.

**In-context learning output:**
The purpose of Go2net Mail is to provide a service for sending and receiving emails. This service allows users to communicate with others through electronic mail, which is a convenient and efficient way to exchange information.

Go2net Mail benefits users in several ways. Firstly, it provides a platform for users to send and receive emails, which is essential for personal and professional communication. Secondly, it allows users to access their emails from anywhere, at any time, as long as they have an internet connection. This makes it easy to stay connected with others, regardless of their location. Thirdly, Go2net Mail provides a secure and reliable way to send and receive emails, which is important for users who need to share sensitive information. Overall, Go2

**Zero-shot learning output:**
Go2net Mail is a free web-based email service provided by Go2net, a popular online directory and search engine. The primary purpose of Go2net Mail is to offer users a reliable and user-friendly email service that allows them to send and receive emails, manage their contacts, and store files online.

Here are some benefits that Go2net Mail offers to its users:

1. **Free Email Service**: Go2net Mail provides a free email service, which means users don't have to pay any subscription fees or maintenance costs.
2. **Web-Based Access**: Users can access their email accounts from anywhere, at any time, using a web browser and an internet connection.
3. **Large Storage Space**: Go2net Mail offers

