# OpenReview forum: "KBLaM: Knowledge Base augmented Language Model"
_ICLR.cc/2025/Conference — ICLR 2025 Poster_

### Official Review · Reviewer_ouM4 · 2024-10-16

**Soundness:** 2
**Presentation:** 3
**Contribution:** 2
**Rating:** 3
**Confidence:** 4

**Summary:**

This paper introduces a new method, KBLaM, for enhancing language models with external knowledge. The main idea can be interpreted as an information compression method of in-context learning (directly putting the entire external corpus in context). Specifically, KBLaM first transforms each piece of knowledge in the knowledge base into continuous key-value vector pairs using a pre-trained sentence encoder. These vector pairs are then mapped into the language model's hidden space via linear adapters. By employing a rectangular attention mechanism, KBLaM enables the integration of external knowledge into the model. Experimental results show that KBLaM achieves performance comparable to in-context learning while offering significantly better scalability.

**Strengths:**

1. Compared to in-context learning, KBLAM introduces linear attention computation scaling relative to the size of the knowledge base (KB). Additionally, it preserves the key advantage of in-context learning by enabling dynamic updates to the knowledge without requiring model fine-tuning.

2. The proposed method also enhances interpretability by potentially offering insights into how the model leverages knowledge through the attention matrix.

3. This paper is clearly-written and easy to follow.

**Weaknesses:**

### 1. Limitations of the method:
- The proposed method encodes each piece of knowledge in the knowledge base separately, without leveraging the graph structure or interconnections between knowledge entities. This could limit the method's ability to capture richer, contextual relationships within the knowledge base.
- The method introduces significant computational overhead by requiring fine-tuning of multiple adapters: a key adapter ($\in \mathbb{R}^{D \times P}$), a value adapter ($\in \mathbb{R}^{D \times P}$), and a query transformation ($\in \mathbb{R}^{D \times D}$) in each transformer layer. For example, using the experimental setup described in the paper, where the hidden sizes of Llama3-8B and the pre-trained OpenAI ada-002 encoder are 4096 and 1536, respectively, the total number of trainable parameters amounts to around **1 billion**—roughly 1/8 of the base model. Compared to in-context learning (training-free), this method represents a substantial increase in resource requirements.
- The performance of the method appears to be heavily dependent on the choice of pre-trained encoder. As shown in Figure 7, using a weaker encoder can lead to a significant drop in performance, in some cases falling below that of the zero-shot baseline.

###  2. Limitations of the experiments:
- The experiments are only conducted on a single base LLM (Llama3-8B). Testing on a wider range of model architectures and sizes would help assess the generalizability of the proposed method.
- The evaluation is based on a small number of datasets, specifically a synthetic knowledge base and Enron. A broader evaluation on more diverse and real-world datasets would provide stronger evidence of the method's effectiveness.
- Missing comparison with Retrieval Augmentation Generation (RAG).

**Questions:**

1. Could you elaborate on the design rationale behind Eq. (9)? Specifically, it seems that the output of each attention layer, $\Tilde{\bf{y}}_n^l$, could alternatively be derived by aggregating the information from both $\Tilde{\bf{v}}_m^l$ and $\bf{v}_i^l$ independently. What specific advantages does the chosen design in Eq. (9) offer over this potential alternative formulation?

2. Could you provide insights on how another competitive baseline, such as RAG, might perform under your experimental settings? Additionally, it would be interesting to know if KBLaM could offer reduced memory overhead compared to RAG.

3. How would KBLaM perform in more challenging KBQA scenarios, such as multi-hop reasoning tasks? Additionally, can KBLaM maintain its advantage over in-context learning in such complex environments?

---

> ### Author Response · Authors · 2024-11-18
> **Author response (1/2)**
>
> We would like to thank the reviewer for the detailed comments, we really find these comments helpful and they indeed help us improve the presentation clarity and provide insights on future directions for improving KBLaM!
>
> Please see our response for each individual points below.
>
> ## Graph structure in the KB
>
> We agree on this limitation. Note that the current synthetic dataset used for training does not contain relationships between the triples.
> In KBLaM, we only considered the underlying *key value* structure of a KB, but we did not utilize the potential graph or hierarchical structure, although we considered this as **a very important future direction**: The research question is *how can we design an attention mechanism that utilizes the graph structure in the context.*
>
> We have included a discussion on this point in the appendix, *Extended future work* section!
>
> ## Resource requirements
>
> Thank you for pointing this out, and we apologize that we were not clear when discussing the computational overhead.
>
> Indeed the total amount of trainable parameters is around 1B. However, we did not find there to be a substantial increase in resource requirements.
>
> Firstly, in the application setting we consider, the training overhead only happens **once** (e.g. in a central server), the learned adapters and query transformations will be distributed to users for serving their own KBs and queries without further retraining or fine-tuning.
>
> Secondly, during inference time, we see the knowledge base as something relatively stable, i.e. many queries will be issued against a single knowledge base. For this reason, we can pre-compute and store all the K and V embeddings, so that the K and V adapters can then be unloaded and would not contribute to any memory usage at inference time.
>
> Lastly, we did not find 1B trainable parameters an issue in our actual training: The training is conducted using one single 80GB GPU with a reasonable batch size of 20 samples.
>
> However, we believe there are a few possibilities to reduce the resource requirement for KBLaM: 1.
> Since the new query transformation is initialized from the pre-trained weights, we can learn an additive low-rank adapter, i.e. LoRA, instead of the full DxD matrices. 2. We can utilize pre-trained LLM's grouped query attention design to reduce the output size of the key and value adapter, e.g. LLaMA only has 8 key and value heads, as such we can let D=1024 instead of 4096.
> We did not consider these strategies in the current version in that we did not find the learnable parameter number an issue at both training and inference time. However, these techniques can be considered in future work when deploying KBLaM in more resource-constrained settings, e.g. on 24GB memory devices.
>
> Again, thank you for raising this and bringing our attention to the parameter size issue, which we did not write about clearly enough in the paper.
>
> ## Performance's dependence on the choice of sentence encoder
>
> You are very correct about this point, and this is a great point that we have touched upon in our ablations studies.
>
> As you point out, the weaker encoder leads to a significant drop in performance. However, this also implies that *with more powerful encoders, we will likely also see better performance!* In fact, as we demonstrate in the third column of Fig.7 in the appendix, if we use the latest and largest embedding model from OpenAI (text-embedding-large), we can outperform the ada-002 version (i.e. the one used in the main text) and shows a performance comparable to ICL.
>
> Furthermore, it is important to note that at inference time, the output size of the embedding model does not affect the LLM's complexity, since knowledge tokens have fixed dimensions regardless of encoder output dimension, as they are passed through the K and V adapters, as stated earlier.
>
> ## Different base LLM
>
> Very good suggestion! We are currently conducting experiments using other language models (in particular, Phi 3 3.8B and Llama 3.2 1B) and will update the results in follow-up comments when ready.
>
>
> ## Design rationale of Eq.9
>
> Thank you for the interesting question! While this would certainly be possible, aggregating $\tilde{v_m^l}$s independently would imply that the post softmax weights for the contribution from the triples values would sum up to one (or some constant if we decide on some relative weighting for the knowledge base part). This can cause issues in our case if the question does not have an answer in the KB, when we expect the model to not attend to the knowledge base, as is shown in Fig.4 (rightmost part), where none of KB triple should be attended over.
>
> Practically speaking, this would also result in the softmax being executed twice, causing additional overhead.

---

> ### Author Response · Authors · 2024-11-18
> **Author response (2/2)**
>
> ## Comparision with RAG
>
> Thank you for suggesting this comparison. We saw in-context-learning as the natural baseline to KBLaM as we include the entire knowledge base in the context, while in RAG, we would be picking items from the context using a separate module.
> However, we have come to appreciate that the retrieval aspect of KBLaM in particular would lead to some interesting comparisons with RAG.
>
> It is hard for us to compare the memory overhead as different RAG systems have different overheads and RAG will most likely be quite cheap in memory, as it would only add a part of the KB to the context, which we also suggest in the manuscript, e.g. in Figure. 1 top.
>
> Performance-wise, we are currently conducting experiments on RAG, and we expect RAG to show answer quality comparable to in-context learning. We will upload the results in a follow-up comment when ready.
>
> Lastly, we would like to emphasize again that "KBLaM beats RAG" (or ICL) is *never* a claim of our paper.
> What we are trying to demonstrate in the submission is the feasibility that
> - External knowledge can be augmented into a pre-trained LLM as fixed-length continuous embedding rather than as discrete tokens;
> - Working with these encoded knowledges, we no longer need an *external retriever*: LLM's attention can serve as an accurate retriever through proper instruction tuning;
>
>
> ## Performance on multi-hop reasoning tasks
>
> Thanks for pointing this setting out! This is an important and exciting problem.
>
> The short answer is: The current version of KBLaM cannot perform multi-hop QA, however, we believe KBLaM, which can see all triples in the context, in theory, can generalize to multi-hop reasoning by incorporating multi-hop QA into the instruction tuning process (which we discussed in the future work section).
>
> To see the reason, assume we have two triples (e1, p1, e2) and (e2, p2, e3) included in KBLaM's context, and we have a query that contains keyword e1 but has e3 as the final answer. KBLaM can answer the query through an implicit two-step reasoning process (similar to chain of thought): KBLaM first retrieves (e1, p1, e2) and generates e2, then retrieves triple (e2, p2, e3) subsequently and generates e3.
>
> We did not include such QAs in the instruction tuning set as for this paper, we wanted to focus on answering the fundamental question of whether incorporating encoded knowledge into the attention layer works at all, and we have found that in the less challenging KBQA settings, it does.
> However, as we mention in the limitations and future work section, this is one of the things that we are quite excited about!

---

> ### Author Response · Authors · 2024-11-21
> **Follow up response**
>
> We would like to provide further details on KBLaM v.s. RAG. We have included a comprehensive discussion of KBLaM v.s. RAG in the general response, where we include additional analysis and evaluations on the latency, memory cost, and overhead of the two methods.
>
> In particular, our evaluation shows that, in a RAG-favorable setting, where we ignore the retriever overhead and include more than 400 triples in KBLaM's context (v.s. 5 triples for RAG), KBLaM is **cheaper** than RAG in terms of both latency and memory!
>
> In terms of answer quality, given 12,800 triples in the KB, RAG shows almost perfect performance since BM25 gives almost 100% top-5 retrieval accuracy in the settings we considered, whereas KBLaM has 94% top-5 accuracy, therefore the answer quality is also correspondingly slightly worse.
>
> Thanks again for suggesting the comparison!
>
> Please let us know if there are any additional concerns or questions we can help address!

---

> ### Author Response · Authors · 2024-11-24
> **Using alternative LLM as backbone**
>
> Hi reviewer ouM4
>
> We have conducted additional experiments per your suggestion
>
> > The experiments are only conducted on a single base LLM (Llama3-8B). Testing on a wider range of model architectures and sizes would help assess the generalizability of the proposed method.
>
> In particular, we have acquired additional results **using an alternative LLM as the backbone**. We applied the KBLaM method, as demonstrated in this paper, to *Phi3-mini-instruct* [1, 2], a 3.8B light scale model. Our results indicate that **KBLaM works equivalently on an alternative LLM** in addition to Llama3-8B used in the paper.
>
> In the two tables below, we report the top-1 and top-5 retrieval accuracy (settings from Fig.5 in the paper) and BERT Score F1 (settings from Fig.6a in the paper) both evaluated on the synthetic KB. We additionally include the Llama results in the table for comparison.
>
> Below are the top-1 and top-5 retrieval accuracy:
>
> | KB size | Phi3 Top 1 $\uparrow$ | Phi3 Top 5 $\uparrow$ | Llama Top 1 $\uparrow$ | Llama Top 5 $\uparrow$ |
> |---------|------------|------------|-------------|-------------|
> | 50      | 0.986      | 1.000      | 0.968       | 0.996       |
> | 100     | 0.965      | 0.999      | 0.978       | 1.000       |
> | 200     | 0.943      | 0.998      | 0.950       | 0.994       |
> | 400     | 0.890      | 0.994      | 0.924       | 1.000       |
> | 800     | 0.854      | 0.989      | 0.924       | 0.998       |
> | 1600    | 0.782      | 0.978      | 0.818       | 0.990       |
> | 3200    | 0.699      | 0.943      | 0.746       | 0.972       |
> | 6400    | 0.614      | 0.895      | 0.664       | 0.944       |
>
>
> Below is the BERT Score F1 (which measures the similarity between the model output and the ground truth answer):
>
> | KB size | Phi3 BERT Score F1 $\uparrow$ | Llama BERT Score F1 $\uparrow$ |
> |---------|-------------------|-------------------|
> | 25      | 0.811             | 0.921             |
> | 50      | 0.818             | 0.914             |
> | 100     | 0.832             | 0.910             |
> | 200     | 0.830             | 0.903             |
> | 400     | 0.820             | 0.891             |
> | 800     | 0.802             | 0.872             |
> | 1600    | 0.782             | 0.846             |
> | 3200    | 0.761             | 0.819             |
> | 6400    | 0.742             | 0.793             |
> | 12800   | 0.728             | 0.771             |
> | 20000   | 0.721             | 0.764             |
>
>
> Note that Phi3-mini version of KBLaM is **comparable with the Llama3-8B version** in terms of retrieval accuracy but the gap in BERT score F1, i.e. the answer quality, is more pronounced. While we have not run experiments for deeper invsetigation, we speculate that there are two potential reasons:
> - Phi3-mini (3.8B) is half the size of Llama3-8B, in particular, the embedding size is smaller than Llama3-8B, which we believe would affect KBLaM's performance. This potentially also indicates that if we have a **larger** base model, KBLaM could show better performance.
> - Due to time constraints, the optimization hyperparameters were directly taken from that of Llama. Potentially some modifications are necessary for optimal performance (e.g. learning rate and training iterations). Nevertheless, such investigation would go beyond the rebuttal timeframe, and we will leave it for future work.
>
> [1] https://arxiv.org/abs/2404.14219
>
> [2] https://huggingface.co/microsoft/Phi-3-mini-4k-instruct
>
> -----------------
>
> Lastly, the public discussion phase will be ending in a few days on November 26th, please let us know if there are any additional concerns we can help address!
>
> Best,
>
> Author of KBLaM

---

> ### Author Response · Authors · 2024-11-25
> **Author follow up**
>
> Dear reviewer ouM4,
>
> We'd like to thank you again for your helpful comments. In addition to the above, we also have results on other datasets, ([PopQA](https://openreview.net/forum?id=aLsMzkTej9&noteId=v8sRTncI7f) and [a perturbed version of our synthetic dataset](https://openreview.net/forum?id=aLsMzkTej9&noteId=fAYi36fowY). We believe we have addressed the weaknesses that you pointed out in your initial review and engaged with the questions.
>
> The public discussion phase will be ending in a few days on November 26th, so please do let us know if there are any additional questions and concerns we can help address!
>
> Best,
> Authors of KBLaM

---

> > ### Comment · Reviewer_ouM4 · 2024-11-28
> >
> > Thanks for the authors’ detailed responses and additional experiments. While your replies address some points I raised and provide helpful insights, I will maintain my evaluation for the following reasons:
> >
> > 1. While I understand your focus on proving the feasibility of embedding knowledge as continuous representations in KBLaM, the lack of integration of graph structure in the KB remains a significant limitation. This design choice limits the richness of the contextual information the model can utilize. While I acknowledge your discussion on future work, this limitation reduces the current method's general applicability, especially for more complex KBs.
> > 2. I appreciate your clarifications about training resource requirements and potential optimizations like LoRA or grouped attention. However, the current method still introduces a significant computational overhead compared to in-context learning, a key baseline. Even though precomputing embeddings helps alleviate inference costs, the high parameter count and adapter complexity limit KBLaM’s practicality in resource-constrained environments. These aspects weigh against the paper's claims of scalability.
> > 3. While I acknowledge that better encoders can enhance KBLaM's performance, the heavy reliance on the encoder's quality adds a dependency that weakens the robustness of the method. This could pose a challenge for researchers or practitioners without access to state-of-the-art encoders.
> > 4. Your explanation of KBLaM’s limitations in handling multi-hop reasoning underscores the method's current narrow focus. While your proposed future direction is promising, this limitation further highlights the gap between KBLaM and more sophisticated KBQA approaches capable of complex reasoning.
> >
> > I commend the clarity of this paper and the proposed method of embedding knowledge as continuous representations. However, the limitations outlined above justify my evaluation scores, particularly in the areas of soundness and contribution. I look forward to seeing future iterations of this work addressing these limitations and expanding the scope of evaluations.

---

> > > ### Author Response · Authors · 2024-11-28
> > > **Author response**
> > >
> > > Dear reviewer ouM4
> > >
> > > We would like to thank you for the follow-up comments and your detailed feedback.
> > >
> > > We fully agree with your observation that the current version of KBlaM cannot handle graph structures in the KB, however, it is worth noting that the research questions we are trying to answer in this project are:
> > >
> > > - Can we augment knowledge as continuous vectors (i.e. a modality) rather than as discrete strings?
> > > - Can we perform retrieval without an external retriever but using LLM's attention only?
> > >
> > > KBLaM demonstrates that these two desirable properties **are possible** through utilizing the structure in the external knowledge, and to the best of our knowledge, this is a novel contribution.
> > >
> > > We believe KBLaM is just *the first paper* for this new regime in knowledge augmentation, therefore we are not aiming to make KBLaM capable of doing everything.
> > > We agree that your suggestions present exciting future avenues. However, we believe current observations, the design choices, the results, and the ablation studies are already thought-provoking and interesting enough to be shared with the community, therefore we decided to write a paper about it.
> > >
> > > > However, the current method still introduces a significant computational overhead compared to in-context learning, ...  the high parameter count and adapter complexity limit KBLaM’s practicality in resource-constrained environments....
> > >
> > > We appreciate your concerns about the computational overhead compared to in-context learning (ICL). We agree that in certain cases, ICL may present lower resource requirements, while in others, we believe KBLaM presents attractive performance benefits.
> > >
> > > **Scarios where ICL requires less resources** In the cases we want to *immediately* use some off-the-shelf LLM, or if the knowledge base will only be used once, we would expect ICL, without the training cost and without an adapter/pre-computed embeddings, to have lower computational requirements.
> > >
> > > **Scarios where KBLaM requires less resources** In settings where we have decided to stick with a particular LLM, the knowledge base remains more stable, with most items not changing between queries, we see KBLaM having an advantage in computation. We'd like to draw your attention to the experiments we've run and posted in the [general response](https://openreview.net/forum?id=aLsMzkTej9&noteId=yTL5KMSwTz).  In particular, if we put a KB of 256 triples into the context, given a query about the KB:
> > >
> > > - ICL takes **2.6** seconds to generate the first token and **45GB** of memory.
> > >
> > > - KBLaM only takes **0.04** seconds to generate the first token and **18GB** of memory. In fact, even with 4096 triples, KBLaM has a 12% faster time to first token than ICL or RAG with **only 5 triples** in the context.
> > >
> > > In terms of the memory requirements of the adapters, at inference time, KBLaM requires an additional query head loaded into the GPU. This **only introduces 6% additional memory overhead**: KBLaM's separate query head has 32 x 4096 * 4096 \~= 540M parameters, which, if stored in bfloat16, takes \~1.08GB of memory, or only 6 percent of the pre-trained 8B model's memory cost (~16GB).
> > >
> > > In terms of the memory overhead of the knowledge itself, the memory overhead from KBLaM with 512 triples in its context is equivalent to the memory overhead in ICL with only 5 triples. For more information on these latency and memory tradeoffs between ICL/RAG and KBLaM, please see the newly added Figure 11 in the appendix.
> > >
> > > In summary, based on our empirical results, we believe that in scenarios where the knowledge base is relatively static, KBLaM significantly reduces the amount of computation required compared to in-context learning, making the method practical in resource-constrained environments.
> > >
> > > > While I acknowledge that better encoders can enhance KBLaM's performance, the heavy reliance on the encoder's quality adds a dependency that weakens the robustness of the method. This could pose a challenge for researchers or practitioners without access to state-of-the-art encoders.
> > >
> > > Thank you for raising the important question of accessibility to state-of-the-art encoders. KBLaM indeed requires OpenAI embedding API. Fortunately, this is **very affordable**, with 1M tokens requiring [$0.05](https://openai.com/api/pricing/) to embed for the model we used, ada v2, when using the batch API. This works out to **$0.01** to embed a KB with **10,000 triples**. For this reason, we do not think this is a significant obstacle to researchers or practitioners. However, we do admit that for researchers in certain regions where OpenAI service is banned, this indeed presents some challenges.
> > >
> > >
> > > We appreciate the constructive feedback and conversation, please let us know if there are any additional questions and concerns that we can address!
> > >
> > > Best Regards,
> > > Authors of KBLaM

---

> > > > ### Author Response · Authors · 2024-12-02
> > > > **Author follow up**
> > > >
> > > > Dear reviewer ouM4
> > > >
> > > > The author reviewer discussion phase is finishing in less than 48 hours, as such we wonder if our follow-up clarifications have resolved your concerns on computational overhead and the accessibility of the encoder model.
> > > >
> > > > Thanks!
> > > >
> > > > Author of KBLaM

---

### Official Review · Reviewer_9hRD · 2024-11-01

**Soundness:** 3
**Presentation:** 4
**Contribution:** 3
**Rating:** 8
**Confidence:** 3

**Summary:**

The authors implement a scheme that encodes Knowledge Base triplets into Key+Value pairs via a pretrained embedding encoder.  The information from the triplets can then be continually 'looked up' by a pretrained local LLM through the training of adapter matrices that (essentially) convert the local LLM representations into embedding space.  Experiments are performed to show that this is effective, including the training of the adapter matrices using synthetic data - which points to greater applicability of the method.

**Strengths:**

The paper presents an interesting new KB embedding lookup technique, with some small-scale experiments.  Also included was some visualisation of the attention across the KB - which was particularly striking for the multistep case (KB items 4+6).

Oeverall, it seems like the ideas in the paper are of general interest, even though they may not be fully fleshed out as presented here.  Even if "KBLaM" isn't the end-point of this line of research, this paper points towards a line of research that could be quite fruitful : Loosely, that weight matrix adapters can be trained to reverse the embedding process using synthetic data.

It also seems that the top-k choices among the Keys are all that are necessary (particularly since the facts are likely stand-alone, rather than mutually reliant), which could mean that much cheaper approximate search could be used to select out relevant entries before doing a full dot-product.

**Weaknesses:**

At several points, the 'linear rather than quadratic scaling' of the method is emphasised.  This seems a weak point, since the size of the tuples $M$ already exceeds the context length $N$, so $MN>N^2$ in this regime (as obliquely mentioned in L284).

The constant $C$ introduced in Eqn 11 feels like something that should have a more principled origin.

small edit: L103 "training samples is not of interests." -> "training samples is not important."

**Questions:**

Looking at the triplet examples (in the Appendix), and the test questions, it seems that there is an extremely high character-overlap between the name-property text and the Questions being asked.  To what extent do the queries have to 'hit' the keys exactly (for instance : Asking about 'Maggie Thatcher' rather than 'Margaret Thatcher')?  Is there an identifiable effect whereby inexact matches get honed-in on subsequent layers/tokens (for instance)?

---

> ### Author Response · Authors · 2024-11-17
> **Author response**
>
> We would like to thank the reviewer for the positive and constructive comment!
>
> First of all, thanks for pointing this out!
> > small edit: L103 "training samples is not of interests." -> "training samples is not important."
>
> We have modified it in the updated manuscript!
>
> Please see below for our responses for each individual point.
>
> ## Using top-k choices.
>
> We are not pruning to top-K choices for the current version, but we agree this would be totally possible, and could potentially make KBLaM more efficient, although there is a potential tradeoff if a question were to involve many triples. We will consider this as a future direction for further improving the efficiency and scalability of KBLaM!
>
> ## Linear v.s. quadratic scaling.
>
> Thank you for pointing out that we were not clear.
> We apologize for the confusion, what are trying to express here is that the overhead of adding knowledge is linear with respect to M for KBLaM, but quadratic **with respect to M** for in-context learning, which flattens and concatenates all triples as part of the prompts. Indeed, we found that in-context learning runs out of memory very quickly as M increases (Fig. 3).
>
> ## Constant C in Eqn 11.
>
> Thanks for pointing this out! The constant C is set as 100, which is the largest number of triples we used during training time.
> We make this decision to ensure that the contribution of the attention over the knowledge to the total output during softmax (Eq. 9) would be of the same scale at test time compared with training time.
> We have modified the manuscript to clarify this point (line 312).
>
> ## Exact keyword hitting
>
> Thank you for suggesting this very interesting setting! Notice that KBLaM performs retrieval by checking the similarity between two embedding vectors, which allows for a certain level of fuzziness in the keywords, e.g. Maggie is very close to Margaret on the embedding space and would not affect the output. This differs KBLaM's retrieval mechanism with keyword-based methods such as BM25.
>
> Following your suggested settings, we considered keeping the triples in the KB unchanged, but perturbing the `name` in the questions, e.g. "What is the objective of Euler's Edifice?" --> "What is the objective of Euler's Building?". We then measure the retrieval accuracy on the perturbed dataset, similar to the setting in Fig.5 without any modification to the model. The results are presented in Fig.10(b) in the appendix. Despite the training data only having contained exact matches, KBLaM can retrieve under such perturbations although there is some degradation in performance compared with the unperturbed setting (Fig.10(a), solid blue lines). BM25, on the other hand, exhibits a much more pronounced performance degradation under question perturbation (Fig.10(a), solid purple lines v.s. Fig.10(b), solid purple lines). As a result, on this perturbed dataset, we see KBLaM consistently outperform BM25 at all KB sizes.
>
> We are also currently evaluating the model's output quality under such perturbation scenarios, and we will update the results in follow-up comments once ready.

---

> ### Author Response · Authors · 2024-11-21
> **Follow up author response**
>
> We have conducted additional experiments for the setting where we *perturbed* certain keywords in the query (e.g. Edifice --> Building, Margaret --> Maggie).
>
> In our previous response, we noticed that **KBLaM retrieves accurately under perturbation** (with little degradation compared with original unperturbed queries).
>
> Our new results indicate that **KBLaM generates answers accurately under perturbation** (with little degradation compared with original unperturbed queries). The BERT F1 score under perturbation is presented below
>
> | KB Size | BERT Score F1 (Clean Queries) | BERT Score F1 (*Perturbed Queries*) |
> |--------:|-------------------------:|----------------------------:|
> | 25 | 0.9209 | 0.8920 |
> | 50 | 0.9136 | 0.8830 |
> | 100 | 0.9104 | 0.8590 |
> | 200 | 0.9033 | 0.8360 |
> | 400 | 0.8910 | 0.8130 |
> | 800 | 0.8718 | 0.7850 |
> | 1600 | 0.8461 | 0.7570 |
> | 3200 | 0.8193 | 0.7350 |
> | 6400 | 0.7928 | 0.7210 |
> | 12800 | 0.7715 | 0.7130 |
> | 20000 | 0.7643 | 0.7110 |
>
> Thanks again for suggesting this interesting setting!
>
> Please let us know if there are any additional concerns or questions we can help address!

---

> > ### Author Response · Authors · 2024-11-22
> > **Author follow up**
> >
> > Dear reviewer 9hRD
> >
> > We would like to sincerely thank you again for this insightful feedback, particularly in recognizing the broader potential KBLaM!
> >
> > The public discussion phase will be ending in a few days on November 26th, as such we would like to know if there are any additional questions and concerns we can help address!
> >
> > Best,
> >
> > Authors of KBLaM

---

> > > ### Comment · Reviewer_9hRD · 2024-11-24
> > >
> > > [Re: Linear v.s. quadratic scaling.]
> > >
> > > Thanks for the clarification - per another reviewer, the key (!) point here is that the
> > > $M$ knowledge items don't need to interact with one another.  Makes sense.
> > >
> > >
> > > [Re: Constant C in Eqn 11.]
> > >
> > > "C is set as 100, which is the largest number of triples we used during training time. " - Yes,
> > > this explanation makes a lot of sense.
> > >
> > >
> > > [Re: Exact keyword hitting]
> > >
> > > Many thanks for running these additional tests for in-exact matches.  Clearly, BM-25 is going to suffer in this case,
> > > but the table you gave doesn't really answer the question about whether purturbing the query has an
> > > impact on performance *per se*.  On the other hand, Figure 10(b) in the Appendix is fairly convincing.
> > >
> > >
> > > [Overall]
> > >
> > > Your feedback was very constructive, and I've nudged up my rating.

---

> > > > ### Author Response · Authors · 2024-11-24
> > > > **Thanks from the author**
> > > >
> > > > Dear reviewer 9hRD
> > > >
> > > > Thank you for increasing the score!
> > > >
> > > > >  but the table you gave doesn't really answer the question about whether purturbing the query has an impact on performance ...
> > > >
> > > > We apologize for the confusion! The goal of the table is to compare KBLaM's **output quality** in ordinal unperturbed queries (second column) v.s. perturbed queries (third column), measured by BERT score F1 (higher the better) that computes the similarity between the model's generation with the ground truth answer.
> > > >
> > > > Best,
> > > >
> > > > Author of KBLaM!

---

### Official Review · Reviewer_W1bY · 2024-11-03

**Soundness:** 3
**Presentation:** 3
**Contribution:** 2
**Rating:** 5
**Confidence:** 4

**Summary:**

This paper proposes a new regime called KBLaM to integrate the entire KB into LLM with a complexity that grows linearly. The paper primarily introduces Rectangular Attention to achieve this. Experimental data shows that the method can be scaled to handle data with up to 10K triples.

**Strengths:**

-	The paper proposes a new method for integrating KB into LLM, and theoretically, this method is one order of magnitude more efficient than the ICL method.
-	It is a neat and interesting way to map each triple into a fixed-length key-value vector pair
-	The main body of the paper is written well, and the explanations for each step of the calculation are precise

**Weaknesses:**

-	The experimental comparison setup is not well-designed:
	No comparisons in the first experiment. How is the retrieval accuracy compares to that of BM25?
	In the last two experiments, although the authors mentioned that ICL encountered memory constraints, it also resulted in fewer data points.
-	The conclusion of the experiment is relatively ordinary. In the last two experiments, the author claims that KBLaM can reason about knowledge and KBLaM knows when it cannot answer a question. However, these two features seem to be the result of the newly proposed structure working in conjunction with the original parameters and structure in LLMs.
-	The novelty of this paper is limited. As the author points out in the appendix, the overall approach is very close to structured prompting[1] and structured attention[2]. If each triple is considered as a demonstration, could the method proposed in this article be seen as transferring structured prompting from text to triples? Additionally, although the author cites work related to structured attention in the appendix, I believe the degree of relevance to these methods is much greater than that to the work on the key-value (KV) cache mechanism and multi-modal language models, and should be cited in the main body.
[1]Hao, Yaru, et al. "Structured prompting: Scaling in-context learning to 1,000 examples." arXiv preprint arXiv:2212.06713 (2022).
[2]Cai, Tianle, et al. "Scaling In-Context Demonstrations with Structured Attention." arXiv preprint arXiv:2307.02690 (2023).
-	There are some repetitive words in the article:
	There is no need to use a large section in Chapter 3 background to introduce Self-attention layer.
	Image label and paper content overlap. For example, “uses the attention score at the 15th layer, averaged over all attention heads, as a classification score for each triple and measures the top-1 and top-5 accuracy” in Fig5 overlaps with the content introduced in the first experiment.
-	The accuracy of ICL on the Enron dataset does not seem to be demonstrated in Fig6(a).

**Questions:**

-	As seen in Fig6, within the range where the ICL method can operate, i.e., less than 200 triples, the ICL method is generally not inferior to KBLaM in terms of accuracy. Can these gaps be narrowed?
-	How does this method compare to retrieval + ICL?
-	How does KBLaM perform on more popular datasets and different types of problems?

---

> ### Author Response · Authors · 2024-11-17
> **Author response (1/2)**
>
> We would like to thank the reviewer for the detailed and constructive suggestions!
>
> Please see below for our response to each point!
>
> ## Comparision of retrieval accuracy with BM25
>
> Thank you for the suggestion of BM25 as a comparison point; we have conducted experiments using BM25, and the results have been interesting!
>
> The experiments have been conducted using the question (e.g. "What is the description of KBLaM?") as the query, and the answer, which includes the name and the property name, as the document (e.g. "The description of KBLaM is a machine learning method that injects knowledge directly into the attention mechanism of LLMs.").
>
> The results are presented in Fig.10 in the appendix.
> We found that BM25 performs favorably, as expected given the large lexical overlap between the query and the document.
> When retrieving from 6400 triples, KBLaM shows a top-5 accuracy of 0.944 on synthetic data and 0.68 on Enron (out-of-distribution), while BM25 achieves a top-5 accuracy of 0.999 on the synthetic data, and 0.96 on Enron
> At the same time, this also demonstrates how Enron presents a more difficult task, even for traditional methods like BM25.
>
> However, under an alternative experiment setting suggested by reviewer 9hRD, where the name of the entity in the question gets perturbed, we find that KBLaM performs better than BM25 in retrieval accuracy at all KB sizes (Fig.10b), despite only having been trained on questions where there was an exact match between the entity name in the question and the triple. We believe the reason is that BM25 is a keyword-matching based approach whereas KBLaM's attention performs retrieval using similarity in the embedding space.
>
> ## Source of KB reasoning capability.
>
> > ...these two features seem to be the result of the newly proposed structure working in conjunction with the original parameters and structure in LLMs...
>
> You are entirely correct; these features are a conjunction of the pre-trained LLM and the adapters introduced by KBLaM: The knowledge comes from KBLaM's encoded triples, and the reasoning ability comes from the pre-trained LLM.
>
> In fact, this is our motivation to keep the pre-trained LLM frozen during instruction tuning: To maximally preserve the base LLM’s reasoning ability and to avoid catastrophic forgetting!
>
> We view this separation of knowledge and reasoning as a key advantage of KBLAM's design, as it allows us to update knowledge independently by injecting knowledge triples at inference time without affecting the model's core reasoning capabilities.
>
> ## Comparision with structured prompting and attention
> We appreciate the reviewer for emphasizing the importance and relevance of structured attention work. As we discussed in the appendix, we also agree that KBLaM is similar to structured prompting and structured attention *in spirit*, in that both achieve more efficient computation by utilizing the independence assumption over certain information in the context. Per your suggestion, we have replaced the multi-modal language model section in the related work section with structured attention.
>
> However, KBLaM is **not** "transferring structured prompting from text to triples", and there exists a couple of differences:
> - Structured prompting presents ICL examples to LLM as discrete tokens, therefore the representations have a size dependent on the length of the ICL examples, making it potentially 20-30 times more expensive in terms of sequence length than the KBLaM strategy, where each triple is represented as a fixed-length continuous vector, similar to the introduction of a new modality, which is why we discuss multi-modal LLM in related work section.
> - Given a triple, structured prompting acquires key and value vectors through LLM's built-in key and value head applied on the same sequence embedding, whereas KBLaM utilizes the structure **inside** each triple and generates the key and value vectors separately from the key and value string of a triple.
> - Structured prompting is equivalent to having a squared attention matrix with sparse block diagonal structure, where each block corresponds to one or more demonstrates, whereas KBLaM's attention matrix is always dense and rectangular because KBLaM does NOT have query vectors for the triples, therefore KBLaM does not require modification of the attention kernel to achieve efficiency, whereas structured prompting would require customized sparsity-aware attention kernel (e.g. using FlexAttention [3]) to actually achieve efficiency.
>
>
> Finally, the motivation is different: KBLaM studies the problem of augmenting LLM with structured external knowledge whereas the structured prompting and attention paper studies the scalability of in-context / many-shot learning problems.
>
>
> [1] Structured prompting: Scaling in-context learning to 1,000 examples.
>
> [2] Scaling In-Context Demonstrations with Structured Attention
>
> [3] https://pytorch.org/blog/flexattention/

---

> > ### Author Response · Authors · 2024-11-17
> > **Author response (2/2)**
> >
> > ## Repetitive words
> >
> > Thanks for pointing this out, the original goal is to make each figure as self-contained as possible, but we agree it introduces too many repetitions.
> >
> > Regarding the introduction to the self-attention layer in Chapter 3, one of the main purposes is to set up the context: The notations and the complexity issues.
> >
> > We are open to the reviewer's suggestion on how to make it more concise!
> >
> > ## Accuracy of ICL on Enron
> >
> > The goal of the figure is to understand KBLaM's performance in an out-of-distribution setting. In-context learning (ICL), on the other hand, does not have a training process therefore it has same performance on both synthetic data and Enron in that Enron is not OOD for it, we therefore choose to clip the y-axis for better visibility of KBLaM's results.
> >
> > But we admit that we are unclear about this point in the original manuscript and would cause confusion, we have now included the additional clarification to Figure 6.(a)'s caption:
> >
> >  > ...In-context learning does not have a training process, showing identical performance on Enron and synthetic data, we omitted its Enron results for graph visibility.
> >
> > ## Comparison and the gap with the ICL method.
> >
> > First of all, we believe that context learning (ICL) is a very strong and competitive approach if judged by the output quality, KBLaM does not yet match it in quality at this moment (note that we are also not claiming nor expecting KBLaM to *outperform* ICL).
> > The issues with ICL are not performance-wise, but: 1. The memory overhead; 2. The lack of interpretability in attention matrix due to causal attention mask.
> >
> > Nevertheless, there do exist possibilities to reduce the gap between KBLaM and ICL's performance through, e.g. using a base encoder of larger capacity: Our results in Fig.7, column 3, demonstrate that using OpenAI's newest and largest embedding model (text-embedding-large), KBLaM shows a much higher BERT F1 score on synthetic data than the ada-002 version, which is the version we used in the main text, i.e. the gap with ICL gets narrowed.
> >
> >
> > ## Comparision with retrieval + ICL
> >
> > Thanks for suggesting this comparison! We believe retrieval + ICL is similar to the Retrieval-Augmented Generation (RAG) setting we discussed in the paper. Indeed this is an important comparison and we are currently performing the experiments, we will update the results in follow-up comments when ready.
> >
> > ## Generalization to other datasets and problems
> > Good point! This is a point also noted by reviewer 6e7T!
> >
> > KBLaM works under the assumption that the external knowledge is already converted into the form of triples, therefore if there are other datasets that store knowledge in triple forms, KBLaM can be generalized to those problems.
> >
> > However, most existing *benchmark QA datasets* (e.g. Natural Questions QA) do not come in the form of triples, we, therefore, consider a setting where we manually convert the dataset's question-answer pairs into knowledge triples and evaluate KBLaM's performance using these derived triples as KB and use the questions as queries. We are currently running experiments under such a setting and will update the results in follow-up comments when ready.

---

> ### Author Response · Authors · 2024-11-21
> **Follow up responses**
>
> We would like to follow up on our previous responses on KBLaM v.s.  retrieval + ICL and dataset/problem generalization
>
> ## KBLaM v.s. Retrieval + ICL
>
> We have posted a general response providing comparisons of KBLaM v.s. Retrieval + ICL (i.e. RAG) from latency, memory cost, and overhead perspectives. In short, our evaluation suggests that, in a RAG-favorable setting, where we ignore the retriever overhead and include more than 400 triples in KBLaM's context (v.s. 5 triples for RAG), KBLaM is still slightly cheaper than RAG in terms of both latency and memory! Accuracy wise, given 12,800 triples in the KB, RAG shows almost perfect performance in that BM25 gives almost 100% top-5 retrieval accuracy in the settings we considered, whereas KBLaM has 94% top-5 accuracy, and the answer quality is also correspondingly slightly worse.
>
> Thank you for encouraging us to conduct the comparison! We agree we lack thorough discussion on this in the initial manuscript.
>
> ## Dataset and problem generalization
>
> We initially attempted to apply KBLaM out-of-box at natural question QA [1] and did not find it very successful.
> A major issue we spot is the dataset does not provide KB triples for KBLaM to operate on, and we found it hard to reverse-engineer the answer string to create triples.
>
> However, on datasets like PopQA [4, 5], we find KBLaM easily applicable: PopQA's setting is identical to KBLaM, where knowledge triples are provided and questions are created using a formatted string based on each triple (Fig.2 in [4]). In this case, we can plug in these triples as KBLaM's KB and use the dataset's questions as queries. Our empirical evaluation shows that, KBLaM trained on synthetic KB can directly perform retrieval, as shown by the top-1 and top-5 accuracy below based on a setting equivalent to Figure.5 from our paper.
>
> | KB size | Top-1 (PopQA) | Top-5 (PopQA) | Top-1 (synthetic KB) | Top-5 (synthetic KB) |
> |----------|--------|--------|---------------------|---------------------|
> | 50 | 0.782 | 0.946 | 0.968 | 0.996 |
> | 100 | 0.710 | 0.901 | 0.978 | 1.000 |
> | 200 | 0.643 | 0.854 | 0.950 | 0.994 |
> | 400 | 0.563 | 0.795 | 0.924 | 1.000 |
> | 800 | 0.466 | 0.713 | 0.924 | 0.998 |
> | 1600 | 0.387 | 0.634 | 0.818 | 0.990 |
> | 3200 | 0.312 | 0.530 | 0.746 | 0.972 |
> | 6400 | 0.246 | 0.454 | 0.664 | 0.944 |
>
> However, we find that KBLaM has difficulties synthesizing good-quality answers, we believe the reason is similar to that of Enron, where the <value>s in the PopQA's triples (a single word) differ significantly from that in the training data (a long sentence).
>
> So from a high level, we believe the answer to "whether KBLaM can be applied to other datasets" is case-by-case:
> - If the dataset's *format* is similar to KBLaM's training data, then KBLaM can be directly generalized. The performance of KBLaM will depend on how similar the dataset is to the synthetic dataset that KBLaM was trained on. Of course, if there is a divergence, the synthetic training dataset should also be updated depending on the intended usecase while applying the same overall methodology. Note that we are introducing KBLaM as a framework for augmenting an LLM with a KB and provide a simple dataset for training such a system rather than providing a final model that can be directly applied to other datasets.
> - If the dataset does not provide triples, then we potentially need to create a separate KB for each dataset where each KB is at least the size of the number of questions we have.
>
> It is also worth noting that the question of **how** to construct a KB effectively goes beyond the scope of KBLaM, with KBLaM focusing on augmenting a pre-trained LLM with knowledge from an existing KB.
>
> [1] Natural Questions: A Benchmark for Question Answering Research, Kwiatkowski et al., 2019
>
> [4] When Not to Trust Language Models: Investigating Effectiveness of Parametric and Non-Parametric Memories, Mallen et al., 2023
>
> [5] https://huggingface.co/datasets/akariasai/PopQA

---

> > ### Author Response · Authors · 2024-11-22
> > **Author follow up**
> >
> > Dear reviewer W1bY
> >
> > We would like to thank you again for your efforts and time for the review. The public discussion phase will be ending in a few days on November 26th. Could you please take a look and let us know if our response is satisfactory?
> >
> > Best, Authors of KBLaM

---

> > > ### Author Response · Authors · 2024-11-28
> > > **Author follow up**
> > >
> > > Dear reviewer W1bY
> > >
> > > We would like to thank you again for your efforts and time for the review. We would appreciate it very much if you could take a look at our response and let us know if it addresses your concerns!
> > >
> > > Best,
> > >
> > > Authors of KBLaM

---

### Official Review · Reviewer_q4dg · 2024-11-04

**Soundness:** 3
**Presentation:** 3
**Contribution:** 2
**Rating:** 5
**Confidence:** 4

**Summary:**

This paper presents KBLAM, a novel approach to augment Large Language Models (LLMs) with external knowledge from a structured knowledge base (KB). The method utilizes pre-trained sentence encoders with linear adapters to transform knowledge into continuous key-value vector pairs, which are then integrated into the LLM via a specialized rectangular attention mechanism. This approach brings the following benefits: improvements in computational efficiency, dynamic knowledge updates, and interpretability compared to existing methods such as Retrieval-Augmented Generation (RAG) and in-context learning.

**Strengths:**

1. The KBLAM method offers a significant advantage in memory efficiency by compressing the kv cache, which enables the model to scale to longer contexts.

2. KBLAM eliminates the need for separate retrieval steps, simplifying the architecture and making the model more efficient compared to RAG.

3. KBLAM allows for runtime KB updates without the need to re-pretrain the model, providing a level of flexibility in deploying LLMs. Additionally, the ability to refuse to answer when the required information is not present in the KB adds a layer of reliability, an essential trait for LLMs used in real-world applications.

4. The paper exhibits a robust methodology with meticulous experimentation. It particularly emphasizes KBLAM’s scalability, efficiency, and interpretability. The use of a rectangular attention mechanism enables the model to scale linearly with the size of the KB. Furthermore, the authors provide comprehensive ablation studies to substantiate their design choices.

**Weaknesses:**

1. I think that the most relevant baseline for comparison with the proposed method should be prompt caching (e.g., as described in the following link). This approach converts the knowledge base into a key-value cache for subsequent use. Compared to this method, I think the advantages of the KBLAM method maybe diminished. For instance: a) it requires instruction tuning; b) the accuracy of retrieval might be substantially lower compared to prompt caching; c) the key-value cache generated by a pretrained encoder is inherently inferior to that obtained from LLMs.
the prompt caching method:
a) https://platform.openai.com/docs/guides/prompt-caching
b) https://ai.google.dev/gemini-api/docs/caching
c) https://proceedings.mlsys.org/paper_files/paper/2024/file/a66caa1703fe34705a4368c3014c1966-Paper-Conference.pdf


2. In the KBLAM method, there is no relative order among different triples, which means there are no position embeddings. In scenarios involving extremely long contexts, this could render the attention mechanism ineffective (the over-smoothing phenomenon), greatly reducing the precision of retrieval. Although the authors propose an attention score scaling method to mitigate this, it would be beneficial to conduct some analytical experiments, such as a needle-in-a-haystack experiment (https://github.com/gkamradt/LLMTest_NeedleInAHaystack), for further analysis.

3. The KBLAM method requires KB instruction tuning. Does this imply freezing the pretrained LLM and only training the adapters? According to the paper, PEFT methods were not used. Consequently, updating the KB kv cache without re-training the LLMs might result in a performance decline, potentially compromising the plug-and-play nature of the method.

4. The performance of the adapter is highly dependent on the data used for instruction tuning. A non-homogeneous kv cache encoder can lead to a significant drop in performance. Therefore, in ablation experiments comparing the pretrained encoder from other sources, it would be best to compare it with the kv cache generated by the prompt caching method, i.e. the LLM itself. This would help readers better understand the performance of the proposed method.

**Questions:**

1. Given that the KBLAM method inevitably leads to information loss (due to the compression of the kv cache and over-smoothing in scenarios with large kb sizes), have the authors considered incorporating a retrieval module to mitigate this issue? Additionally, it might be beneficial to use retrieval scores as a position bias (for example, the kv cache closer to the prompt corresponds to a higher relevance of the triple, which aligns with the inherent characteristics of the attention mechanism and should effectively enhance the method's performance).

2. Have the authors experimented with making the method more modular, such as by using PEFT (Parameter-Efficient Fine-Tuning) methods and training only the adapters part? If so, what were the results of this approach?

---

> ### Author Response · Authors · 2024-11-16
> **Author response (1/2)**
>
> We thank the reviewer for their thoughtful feedback.
>
> Firstly, we would like to clarify that KBLAM is **not** a KV cache compression method. Rather, it is an approach for encoding structured knowledge triples into continuous representations (knowledge tokens) that can be directly integrated with pre-trained LLMs.
>
> ## Relationship with prompt caching
>
> Thanks for pointing out this very important and related class of methods. Indeed prompt caching shares some similarities with KBLaM: Both approaches involve pre-computing and storing some key and value vectors for subsequent use. In particular, we extremely appreciate the reviewer for pointing out the Prompt Cache paper [1], and we believe that the memory management techniques can potentially improve KBLaM's efficiency in deployment. It is also worth noting that OpenAI's prompt cache was introduced on Oct 01 [3], Claude's prompt caching API was introduced on Aug 14 [4], and it is wonderful to see concurrent work with the development of KBLaM on the timely and important issue of LLM deployment. We have included a new paragraph in the extended related work section to discuss prompt caching.
>
> However, KBLaM is **different** in four ways from the existing prompt caching mechanism. Consider a prompt caching strategy where we generate and cache the KV vectors for the string triple_1, triple_2, ..., triple_M, i.e. the in-context learning setting in our manuscript. In this case, KBLaM differs from prompt caching in three aspects:
>
> 1. **Construction of KV vectors**: KBLaM encodes the triple's <name> and <property> as the key vector, and <value> as the value vector, so the resulting key and value vectors only introduce the overhead of 1 token. Prompt caching directly presents triples as discrete tokens to LLM, generating keys and values from the set of tokens, and would cost ~20-30 times additional overhead in token count due to the length of each triple.
>
> 2. **Interpretability of the attention mask**: KBLaM shows interpretable attention matrices as shown by Fig.4 in the manuscript. It is unclear if prompt caching mechanism is capable of providing a similar effect. We checked [1] thoroughly and did not find relevant experiments or discussion on retrieval behavior/performance.
>
> 3. **Reusability and dynamic update of key and value vectors**: Perhaps a more severe issue of the prompt caching strategy is that, if we modify a single word of triple_1, then the stored cache would technically become **invalid** because the cache of all other triples depends on the content of triple_1 due to causal self-attention mask, which is also an issue suggested by Claude's official document [2]. However, KBLaM generates the knowledge token for each triple **independently**, therefore modifying a single triple would NOT affect the stored KVs of other triples, allowing dynamic updates of external knowledge with minimal overhead.
>
> Lastly, we consider modifying the attention mask such that each triple *cannot see each other* to mimic the "modular prompt cache" strategy described in [1] per our understanding. However, we find that such attention masks do not work with our setting, the model starts to mix all triples' contents together when answering questions. In fact, we suspect that incorporating a big number (e.g. 100-1,000) of independently encoded KV cache into the context of a pre-trained LLM is non-trivial since it is equivalent to having an *extremely sparse* block diagonal attention matrix that differs significantly from the dense attention matrix seen at the pre-training time.
>
> Such failure pattern is also observed by a concurrent ICLR submission [5], which states in line 367: *It is not advisable to directly switch from self-attention to Block-Attention, as it will lead to a sharp drop in the accuracy of the model.* and spends the whole paper studying how to fine-tune an LLM to adapt to such pattern.
> We believe getting such attention regime to work in our setting is non-trivial and beyond the scope of the current paper, we will leave deeper investigation as future work (which is a super interesting problem!). We are also open to the reviewer's suggestions on how to perform comparisons with prompt caching.
>
> [1] Gim, In, et al. "Prompt Cache: Modular Attention Reuse for Low-Latency Inference.(2023)." arXiv preprint cs.CL/2311.04934 (2023).
>
> [2] https://docs.anthropic.com/en/docs/build-with-claude/prompt-caching#what-can-break-the-cache
>
> [3] https://openai.com/index/api-prompt-caching/
>
> [4] https://www.anthropic.com/news/prompt-caching
>
> [5] https://openreview.net/forum?id=7zNYY1E2fq

---

> ### Author Response · Authors · 2024-11-16
> **Author response (2/2)**
>
> ## Relative ordering of triples
>
> Notice that we choose not to introduce relative ordering **by design**. The data structure underlying all triples can be understood as a set, an orderless data structure, and our rectangular attention computation (Eq. 9) guarantees that the order of the knowledge tokens does NOT affect the output.
>
> In fact, we believe this is a potential advantage over the prompt cache mechanism, as the prompt cache mechanism would require careful engineering and modification to the positional encoding, which is not required in KBLaM's case since all the KV pairs are **orderless**
>
> Notice also that the lost in the middle / positional bias in long-context LLM is mainly caused by the usage of positional encoding and causal attention mask, both do not exist in KBLaM's knowledge tokens, therefore it is unclear if such issues apply to KBLaM's rectangular attention. Regarding the oversmoothing issue, we are unclear about the exact definition of this problem. Could you potentially share a reference so we can learn more about this and potentially add this to the paper?
>
> With that said, there does exist absolute distance information between each knowledge tokens' key and prompt/question part's query (which is the same for every knowledge triple), which we believe can be captured by the key adapter implicitly, therefore we choose not to further explicitly include a global positional encoding for the knowledge tokens.
>
> ## Training and modularity of KBLaM
>
> In KBLaM, we **freeze** the pre-trained model's weights. We only optimize the linear adapters and separate query heads at each layer specifically for looking up the keys of the knowledge tokens, all the learnable parameters are **separated** from the pre-trained LLM's weights.
>
> In fact, KBLaM's implementation is already modular (as can be seen in the supplementary materials, e.g. in `model_utils.py`). The pseudo-code snippet below is an example of how KBLaM is used in practice
> ```python
> model = KblamForCausalLM.from_pretrained(DIR_TO_PRETRAINED_LLAMA_WEIGHT)
> model.load_query_head(DIR_TO_LEARNED_QUERY_HEAD)
> my_kb = {...}
> kb_kv = [KBEncoder.encode(triple) for triple in my_kb]
>
> # Answer the question using the KB
> model.generate(
>     input_ids, # Tokenized question about the KB
>     kb_kv=kb_kv
> )
>
> # If we set kb_kv as None, we fall back to pre-trained model
> # i.e. set M=0 in Eq.9, we answer the question using LLM's internal knowledge
> model.generate(
>     input_ids,
>     kb_kv=None
> )
> ```
>
> We appreciate the reviewer for pointing this out, we believe that modularity of an approach is indeed a very important property for practical usability and we would appreciate suggestions from the reviewer on how to further improve the modularity of KBLaM.
>
> ## Using LLM's built-in KV cache instead of sentence encoder
>
> Thanks for pointing this approach out. First of all, such an approach would result in ~20-30 times additional memory overhead in that now each triple would no longer be represented by a fixed length key value vector pair. However, this could potentially give better performance thanks to the larger capacity and homogeneity, as the reviewer suggests. With that said, as stated in the prompt caching part, we still believe non-trivial adaptation and fine-tuning would be required to make such approaches work, we consider this as one of our future improvement directions for KBLaM: Using LLM's built-in embeddings as the base embedding (Eq. 4 in the manuscript) as an alternative to the ones generated from sentence encoder.

---

> > ### Comment · Reviewer_q4dg · 2024-11-22
> > **Thank you for the authors' response**
> >
> > Thank you for the authors' feedback and clarification, which addressed some of my concerns. However, I would like to further clarify my questions:
> >
> > 1. I understand the distinction between the prompt caching method and the current approach. I would like to see a comparative analysis of the current method against similar methods since they do not require additional training and could serve as an easily implementable baseline. Could the authors provide a simple comparison (for example, by encoding the knowledge base into a key-value cache and reusing it)?
> >
> > 2. Regarding Weakness 2, the issue of over-smoothing refers to the phenomenon where, with extremely long contexts, the large number of tokens results in attention scores becoming nearly uniform after applying softmax. This hinders the retrieval of important information. I highlighted this in Weakness 2 because the KBLaM method is orderless, which inherently leads to over-smoothing. This often results in significant retrieval issues, and I believe it is necessary to conduct a "needle-in-a-haystack" experiment to illustrate this.
> >
> > 3. As for Weakness 3, I appreciate the authors' clarification. Since the adaptor requires training, the KBLaM method indeed incurs some overhead compared to the prompt caching method. I understand this is intended to reduce the storage cost of key-value vectors, and I suggest the authors highlight this aspect in their analysis within the paper.
> >
> > 4. As pointed out in Weakness 4, I believe it is still necessary to conduct a performance comparison with key-value caching methods (such as key-value caching or prompt caching) to demonstrate the advantages and disadvantages of the current method relative to these approaches.
> >
> > 5. Regarding Question 1, I believe that the retrieval module could significantly enhance the current method, and I encourage the authors to provide a brief analysis on this matter.

---

> ### Author Response · Authors · 2024-11-22
> **Author follow up**
>
> Dear reviewer q4dg
>
> We would like to thank again for your efforts and time for the review. The public discussion phase will be ending in a few days on November 26th, and we think we have addressed the points regarding prompt cache and modularity raised in the review.
>
> Could you please take a look and let us know if our response is satisfactory?
>
> Best,
> Authors of KBLaM

---

> ### Author Response · Authors · 2024-11-22
> **Author follow up response (1/2)**
>
> We would like to thank the reviewer for the updated feedback.
>
> Please see below for our response to the updated feedback!
>
> # KBLaM v.s. prompt caching
>
> Thank you for the clarification. First of all, we believe there are two types of prompt caching strategies:
>
> Approach 1. We generate the KV cache for each triple **independently** and directly plug in all these KV cache into the context. This strategy is similiar to KBLaM's approach. However, our internal experiments as well as a concurrent ICLR submission [5] have found that this strategy would break the model's output without fine-tuning. In short: it seems one cannot take an LLM pre-trained with a dense causal attention mask and turn the mask into an extremely sparse block structure without fine-tuning. While this highly structured attention mask would be exciting future work, we do not believe there currently exists a good baseline to compare KBLaM against for this approach.
>
> Approach 2. We can concatenate all the triples as a single long string and store its KV cache for future use. This approach, which is more widely adopted, and seems to be the approach used by OpenAI and Anthropic given their cache invalidation conditions, could be applied to this problem. As the entire KV cache is kept in this case, the characteristics are similar to in-context learning, for which we already have results, so we can already compare its performance with KBLaM:
> - **Answer quality**: prompt caching will be **identical** to the in-context learning methods we considered in the paper, i.e. slightly better than KBLaM on in-distribution data (see figure 6a).
> - **Latency**: During the prefilling stage (i.e. when generating the KV cache), the prompt cache will be **much more expensive** than KBLaM in that it needs to feed a very long string through the transformer (through self-attention and all MLPs) whereas KBLaM only requires a pre-trained sentence encoder + linear adapter. During inference time, it will be slightly slower than KBLaM in that the length of the KVs is kb_size * triple_length, whereas KBLaM's KVs are only of length kb_size.
> - **Memory (prefilling)**: During the prefilling stage (i.e. when generating the KV cache), prompt cache requires **a quadratic memory overhead**. The amount of memory required would be equivalent to that required by ICL, and we found that this does not fit in the 80GB A100 memory once more than 200 triples are presented in the context (as demonstrated by Fig.3 in the manuscript), unless more sophisticated techniques such as PagedAttention or RingAttention are used. KBLaM on the other hand, does not require a LLM to generate the KVs, in fact, the encoding process can potentially be done on CPU only, as it's just an OpenAI API call + linear transformations.
> - **Memory (inference)**: During the inference stage, a prompt cache will have an attention matrix of shape (question_length, kb_size * triple_length) whereas KBLaM's is of shape (question_length, kb_size). Therefore we suspect that prompt cache will incur a memory overhead **~20 times larger** then KBLaM.
> - **Interpretability**: We are unaware of any interpretability from the attention matrix under prompt cache / in-context learning, while KBLaM demonstrates highly interpretable structure (Fig. 4 in the manuscript).
> - **KV reusability**: Notice that we focus on the KV cache reusability a lot because we assume the KB contents will be updated and modified in a daily manner. Assume one triple in a KB of $M$ triples is modified, KBLaM's update overhead is $O(1)$ since only a single KV vector pair will need to be changed by design. Prompt caching's overhead will technically be $O(M^2)$ since all the KV cache will need to be re-computed due to the autoregressive nature of the causal attention mask.
>
> If one does not believe the interpretability and KV cache reusability are issues, one can argue that the memory and latency issue of generating KV cache only happens **once**, so the overhead is acceptable. *However*, consider a setting where the KB is generated/consumed on a low-power device, e.g. on a laptop (in fact this is an application of KBLaM we are pushing). With KBLaM, the overhead of generating KVs would be a single OpenAI embedding API call (or a small sentence BERT) + a linear transformation. Additionally, the KVs for each triple, by design, can be generated independently, so if the KB is too large to fit into a batch, we can simply use a single for-loop to encode the triples sequentially. However, generating KVs for the prompt cache strategy will be much more challenging on such devices without cutting-edge GPUs.
>
> [5] https://openreview.net/forum?id=7zNYY1E2fq

---

> ### Author Response · Authors · 2024-11-22
> **Author follow up response (2/2)**
>
> ## Issue of over-smoothing
>
> Thank you for clarifying the over-smoothing phenomenon. While we do find some drop in performance as we increase the number of triples, as shown in Figure 5, KBLaM learns to attend/retrieve to the correct item with high accuracy, with **>90% top-5 accuracy even with 6400 triples** (which, if flattened as string, takes > 100K tokens). We also did not observe any over-smooth issues when inspecting the attention matrix. Also notice that our retrieval accuracy is directly computed through the post-softmax attention score, we suspect a severe accuracy drop would occur if over-smoothing happens.
>
> In addition, we consider our experiment setting, with many irrelevant triples in the context and just one correct triple, to be extremely similiar to the needle-in-a-haystack setting.
> This retrieval accuracy results in high-quality responses even with up to 20,000 entities, as shown in Figure 6. Here, we show that the output is semantically similar to the ground truth, even with many input triples (0.76 BERT F1 score with 20,000 triples). We also show that the model can correctly apologize when the required "needle", or triple, is not present in the context, and the performance with the number of triples is illustrated in Fig. 6(c).
>
> ## KBLaM needs adapter training
>
> Indeed, KBLaM requires the training of additional adapters, however, this overhead only needs to **occur once**, (e.g. in a central server), the adapters can later be distributed to users for serving their own KBs and queries without further retraining or fine-tuning: The goal of KBLaM's adapter training process is to learn *a linear mapping* from sentence encoder's space into a pre-trained LLM's semantic space, **not** to memorize factual knowledge into weights.
> It is perhaps also worth noting that the adapter training only requires 24-48 hours on a single 80GB gpus.
>
> Nevertheless, we do agree that KBLaM requires some training, unlike in-context learning / prompt caching, we have included additional discussion on this matter per your suggestion in Appendix.D, the extended future work and limitation section.
>
> ## Usage of external retriever
>
> Notice that one of the major contributions and motivations of KBLaM is to answer the research question:
>
> *Can we perform retrieval **without using an external retriever** but solely using the attention itself?*
>
> And our answer is: Yes. Under the settings considered, LLM's attention indeed demonstrates highly interpretable retriever behavior (Fig. 4)
>
> Indeed, we agree that including an external retrieval system in the pipeline can improve the performance, but this would go against the motivation of our research goal.

---

> > ### Author Response · Authors · 2024-11-26
> > **Author follow up**
> >
> > Dear reviewer q4dg
> >
> > We appreciate your time and effort in writing the detailed review: It indeed helps us to build a deeper understanding regarding the differences between KBLaM v.s. prompt caching!
> >
> > It would be very nice if you could let us know if our arguments look sensible and convincing to you! Any additional feedback is appreciated!
> >
> > Thanks,
> >
> > Authors of KBLaM

---

> ### Author Response · Authors · 2024-12-02
> **Author follow up**
>
> Dear reviewer q4dg
>
> We appreciate your time and effort in writing the detailed review: The discussion period is ending soon in less than 48 hours, therefore, it would be very nice if you could let us know if our comparison between KBLaM and prompt caching looks sensible and convincing to you!
>
> Thanks,
>
> Authors of KBLaM

---

> ### Comment · Reviewer_q4dg · 2024-12-02
>
> Thank you for the detailed responses and clarifications regarding one of the key claims of the paper, "Can we perform retrieval without an external retriever, using only the attention mechanisms of LLMs?". After reviewing the claim and previous discussions, I decide to maintain my current evaluation, due to the following points:
>
> 1. Utilizing the LLM's attention mechanism for KB retrieval can potentially suffer from the following issues compared to existing RAG methods:
>    - a) The efficiency and effectiveness may not necessarily outperform traditional retrieval methods. As the author mentioned in other comments, for exact match retrieval, the accuracy of BM25 can be higher than KBLaM; for fuzzy retrieval or complex question-answering scenarios, the authors can leverage sentence encoders, combining with vector retrieval tools like Faiss or other commonly used KBQA retrieval methods, which could be better choices, especially in the low-resource scenarios emphasized by the authors.
>    - b) The lack of positional encoding in the attention mechanism, i.e., the orderless setting described in the paper, is essentially equivalent to using a vector retrieval module for content retrieval. However, since the retrieval module is the LLM itself (the query heads), the retrieval cost may not have an advantage over the vector retrieval approachs.
>
> 2. Compared to the prompt caching/RAG techniques, the current approach still has significant drawbacks according to the previous discussions:
>    - a) The non-homogeneous kv_cache will naturally underperform prompt caching/RAG, and it requires additional training overhead, especially when the sentence encoder and LLM combination changes.
>    - b) When evaluating the kv_cache reusability, the author did not consider the retrieval advantages of prompt caching/RAG. We do not need to concatenate all the KB triples into a complete content for kv_cache generation; instead, we can retain some commonly used triple combinations or individual triples (just like the LRU caching mechanism). In this case, even with content updates, the update and storage cost would be relatively lower and acceptable.
>
> While the above points constrain me from assigning a more favorable rating, I still recognize that the author's KBLaM method can significantly reduce the required kv_cache size, which represents a beneficial feature. The author's approach can be viewed as an integration of the vector retrieval module within LLMs, which could mitigate the information loss during the retrieval step. However, this approach may incur a higher cost compared to conventional retrieval methods. It is suggested that the author consider incorporating existing prompt caching, RAG, retrieval, and related techniques to further enhance the current method, without being constrained to utilizing the attention mechanism for all the components. This could result in a more broadly applicable and higher-quality knowledge-base-enhanced LLM application framework.

---

> ### Author Response · Authors · 2024-12-02
> **Author follow up response**
>
> Thank you again for your continued engagement! While we agree with many of your points, we wanted to make a few clarifications.
>
> ## Attention as retriever v.s. traditional retrieval methods
>
> > The efficiency and effectiveness may not necessarily outperform traditional retrieval methods.
>
> Yes, this is true, in fact, we **never claim** KBLaM's retriever a better one in our submission nor do we expect this to be true. As we mentioned earlier, traditional retrieval methods are backed by decades of research from the information retrieval community, so we expect all these traditional methods to be extremely robust and powerful.
>
> Also, we would like to clarify that KBLaM's attention is **never explicitly used as a retriever** to select triple during generation. We study the retrieval performance solely to demonstrate that our attention accurately (Figure 5) and interpretably (Figure 4) attends to the correct piece of knowledge. As we visualized in Fig.1, a user query is answered directly through **one single forward propagation** of a transformer without utilizing external modules.
>
> Our main comparison is with in-context learning (which is also the baseline we considered): Recall that a LLM's query and key mechanism look *extremely similar to a retrieval system*. So why can't we look at the attention matrix of in-context learning to figure out which triple in the KB the model is utilizing for generating the output? The answer comes in two folds
> 1. The self-attention entangles the representation of all triples together.
> 2. Each triple is represented by multiple tokens so it is hard to locate the most representative key embedding for inspection.
>
> KBLaM, through rectangular attention and a fixed-length representation of each triple resolves both issues (of course, we also gain much better scalability).
> In particular, we show that the favorable retriever behavior in attention arises with pure instruction tuning using paired data, without any explicit retrieval objective. The "attention retriever" shows fairly accurate performance comparable to a baseline of BM25.
>
> ## Comparision with prompt caching
>
> We agree that combining retrieval with KV cache is an intuitive, natural, and promising idea. However, we are not aware of any published work that considers this approach. The only works we are aware of in this direction are two concurrent ICLR submissions [1, 2].
>
> *We kindly ask the reviewer to point us toward some existing related works since we are very interested to see how to make such a system work in practice.*
>
> In particular, we expect this strategy requires some non-trivial modifications upon the naive prompt caching strategy seemingly employed in commercial settings:
> - As you suggest, we can retain some *commonly used triple combinations*, however, this would involve designing a dedicated caching algorithm/system for estimating the triple combination frequencies.
> - If we consider storing the KV cache for one triple only, then in case the query involves more than one triple or no triple in the KB, it is unclear how a system can handle these cases, especially as the number of triples involved in the query gets higher, as the self-attention *between* these triples will not be in such a modular KV cache (whereas KBLaM and ICL can both handle such cases).
> - As studied in [1, 2], making sure the positional encoding works properly is non-trivial. (However KBLaM does not have such issue since all triples are orderless)
>
> Nevertheless, we are glad that the reviewer acknowledged the kv cache reusability as an issue, and we believe the ideas provided by the reviewer would make a very exciting future paper!
>
> [1] https://openreview.net/forum?id=7zNYY1E2fq
> [2] https://openreview.net/forum?id=x7NbaU8RSU

---

### Official Review · Reviewer_6e7T · 2024-11-05

**Soundness:** 3
**Presentation:** 3
**Contribution:** 3
**Rating:** 8
**Confidence:** 3

**Summary:**

This paper focuses on the open-domain question-answering task and proposes the Knowledge Base Augmented Language Model (KBLaM) framework. Specifically, KBLaM augments a large language model (LLM) with an external knowledge base (KB) at inference time by first transforming the knowledge in the KB into continuous key-value vectors using pre-trained encoders, then integrating them into the LLM through a specifically designed rectangular attention mechanism. Unlike Retrieval-Augmented Generation (RAG), the proposed KBLaM does not require costly retrieval, and, unlike in-context learning (ICL), KBLaM achieves comparable performance with significantly lower computational overhead. Experiments on both a synthesized KB and a real KB dataset demonstrate the effectiveness of the proposed method.

**Strengths:**

1. The motivation of the paper is strong. It addresses scenarios where external knowledge is required for LLMs to solve tasks and explores how to integrate that information into LLMs efficiently and effectively. This paper identifies the limitations of current methods, including long in-context learning (ICL) and retrieval-augmented generation (RAG), and proposes a new solution that mitigates these limitations.
2. The proposed KBLaM, to my knowledge, is novel and offers the following advantages:
- It transforms external unstructured knowledge into a structured KB with knowledge triplets, which compresses lengthy context information and facilitates convenient integration of knowledge into LLMs.
- It encodes knowledge triplets as key-value vectors of uniform size, enabling straightforward incorporation into LLMs. The structured nature of the triplets also allows efficient retrieval of relevant knowledge within the KB.
- Each knowledge triplet can be encoded and incorporated into LLMs independently, providing greater scalability and lower complexity than ICL. This approach also offers flexibility for updating the knowledge within the KB.
- It also allows high interpretability via viewing the attention matrix.
3. Overall, this paper provides solid evidence through extensive experiments and ablation studies to demonstrate the effectiveness of the proposed method. The writing is also clear and easy to follow.

**Weaknesses:**

1. While this paper includes experiments with both synthetic and real-world KB datasets, the baselines are limited to only ICL and zero-shot approaches. Recent state-of-the-art methods, such as RAG-based and long-context models, are not included as baselines, making it difficult to assess the effectiveness of the proposed method relative to these approaches. Including results from these comparisons would strengthen the evaluation.
2. The generalization of KBLaM is less well-known. While it has been tested on an OOD dataset, other commonly used QA datasets have not been explored, such as [1][2][3][4]. Please see the questions below.

---
[1] Natural Questions: A Benchmark for Question Answering Research, Kwiatkowski et al., 2019

[2] TriviaQA: A Large Scale Distantly Supervised Challenge Dataset for Reading Comprehension, Joshi et al., 2019

[3] MuSiQue: Multihop Questions via Single-hop Question Composition, Trivedi et al., 2022

[4] When Not to Trust Language Models: Investigating Effectiveness of Parametric and Non-Parametric Memories, Mallen et al., 2023

**Questions:**

1. In Figure 6(a), why are the in-context learning results (orange line) missing from the Enron dataset?
2. I’m interested in the generalization of the proposed method. For other QA datasets and benchmarks commonly used by the community, such as [1][2][3][4], is it possible to extend KBLaM to address these problems? How can a KB be constructed for them? Do you need to build a KB for each dataset? What is the size (number of knowledge triplets) of the KB that is sufficient or suitable to solve these problems?

---

> ### Author Response · Authors · 2024-11-16
> **Author response**
>
> We would like to thank the reviewer for the constructive comments!
>
> ## Comparison with RAG and long context
>
> We are currently conducting comparisons with RAG, namely using a BM25 as the retriever to load relevant triples in the context, and we will update the results in follow-up comments when ready.
>
> Note that we are not expecting KBLaM to outperform RAG in performance on the Q&A tasks we considered in its current form.
> Instead, we believe KBLaM's main advantage over RAG lies in the simplicity of the pipeline, with knowledge integrated directly into the attention, enabling the ability to process the entire knowledge base in context, enabling features like answer refusal.
>
> Notice that the in-context learning baseline is equivalent to the long-context method, where all triples are flattened and concatenated together to form a long prompt. However, the maximum number of triples we can include in the context is 200 due to memory constraints (Figure. 3 orange line. 200 triples here corresponds to approximately 6000 tokens). If we have extra memory, we predict that in context learning's performance will continue to drop as the number of triples increases.
>
> ## In-context learning results on Enron dataset.
>
> The goal of the Enron part of figure 6(a) is to understand KBLaM's performance in an out-of-distribution setting. In-context learning (ICL), on the other hand, does not have a training process therefore it has the same performance on both Synthetic data and Enron in that Enron is not OOD for it, we therefore choose to clip the y-axis for better visibility of KBLaM's results.
>
> We admit that this would cause confusion and lacks clarity, we have now included the additional clarification to Figure 6.(a)'s caption:
>
>  > ...In-context learning does not have a training process, showing identical performance on Enron and synthetic data, we omitted its Enron results for graph visibility.
>
> ## Generalization of KBLaM to other QA datasets
>
> Thank you for pointing out the lack of experiments on traditional Q&A datasets. We acknowledge two key considerations regarding traditional QA datasets:
>
> Firstly, many knowledge sources and datasets (e.g. wikipedia) are highly likely to have been included in the LLM's pre-training data, making it difficult to differentiate between answers derived from provided knowledge triples versus the model's parametric knowledge. Secondly, most datasets do not present knowledge in the form of triples, a prerequisite condition KBLaM assumes.
>
> Nevertheless, it is still possible to test KBLaM on these datasets by converting question-answer pairs into knowledge triples and evaluating KBLaM's performance using these derived triples as KB and using the questions as queries.
>
> We are currently running experiments under this setup using the NaturalQA dataset and will update the results in follow-up comments.

---

> > ### Comment · Reviewer_6e7T · 2024-11-18
> > **Thanks for authors' response**
> >
> > I appreciate the authors' response and their commitment to the revision.
> >
> > >Note that we are not expecting KBLaM to outperform RAG in performance on the Q&A tasks we considered in its current form. Instead, we believe KBLaM's main advantage over RAG lies in the simplicity of the pipeline, with knowledge integrated directly into the attention, enabling the ability to process the entire knowledge base in context, enabling features like answer refusal.
> >
> > I acknowledge the potential benefits that KBLaM may offer. However, it would be helpful to see its performance in terms of generation accuracy and costs (such as memory, latency, etc.) compared to RAG, so that people can have an overall understanding of these two methods.
> >
> > >Firstly, many knowledge sources and datasets (e.g. wikipedia) are highly likely to have been included in the LLM's pre-training data, making it difficult to differentiate between answers derived from provided knowledge triples versus the model's parametric knowledge. Secondly, most datasets do not present knowledge in the form of triples, a prerequisite condition KBLaM assumes.
> >
> > In terms of this point, [1] specifically addresses this challenge and creates a dataset designed for it. It might also be easy to convert the knowledge into triplets.
> >
> > [1] ToolQA: A Dataset for LLM Question Answering with External Tools

---

> ### Author Response · Authors · 2024-11-21
> **Follow up author response**
>
> Thank you again for the replies and suggestions! Please see below for our follow-up responses
>
> ## KBLaM v.s. RAG
>
> We completely agree with you on the importance of a comprehensive comparison of KBLaM v.s. RAG, and we have posted a general response providing comparisons of latency, memory cost, and overhead. In short, our evaluation suggests that, in a RAG-favorable setting, where we ignore the retriever overhead and include more than 400 triples in KBLaM's context (v.s. 5 triples for RAG), KBLaM is still slightly *cheaper* than RAG in terms of both latency and memory! Accuracy wise, given 12,800 triples in the KB, RAG shows almost perfect performance in that BM25 gives almost 100% top-5 retrieval accuracy in the settings we considered, whereas KBLaM has 94% top-5 accuracy, and the answer quality is also correspondingly slightly worse.
>
> Thank you for encouraging us to conduct the comparison, we have acquired a deeper understanding of the pros and cons of KBLaM and RAG through this process!
>
> ## Dataset generalization
>
> After some initial attempts at natural question QA [1], we found that, out of the box, the KBLaM model we have currently trained cannot be directly applied to this setting. A major issue we spot is the dataset does not provide KB triples for KBLaM to operate on, and we found it hard to reverse-engineer the answer string to create triples.
>
> Thank you for the suggestions of additional datasets! While ToolQA also has the limitation of not having triplets, we found PopQA [4, 5], a dataset where KBLaM is definitely applicable: PopQA's setting is identical to KBLaM, where knowledge triples are provided and questions are created using a formatted string based on each triple (Fig.2 in [4]). In this case, we can plug in these triples as KBLaM's KB and use the dataset's questions as queries. In fact, according to our empirical evaluation, KBLaM trained on synthetic KB can directly perform retrieval, as shown by the top-1 and top-5 accuracy below based on a setting equivalent to Fig. 5 in our manuscript:
>
> | KB size | Top-1 (PopQA) | Top-5 (PopQA) | Top-1 (synthetic KB) | Top-5 (synthetic KB) |
> |----------|--------|--------|---------------------|---------------------|
> | 50 | 0.782 | 0.946 | 0.968 | 0.996 |
> | 100 | 0.710 | 0.901 | 0.978 | 1.000 |
> | 200 | 0.643 | 0.854 | 0.950 | 0.994 |
> | 400 | 0.563 | 0.795 | 0.924 | 1.000 |
> | 800 | 0.466 | 0.713 | 0.924 | 0.998 |
> | 1600 | 0.387 | 0.634 | 0.818 | 0.990 |
> | 3200 | 0.312 | 0.530 | 0.746 | 0.972 |
> | 6400 | 0.246 | 0.454 | 0.664 | 0.944 |
>
> However, we find that KBLaM has difficulties synthesizing good-quality answers, we believe the reason is similar to that of Enron, where the <value>s in the PopQA's triples (a single word) differ significantly from that in the training data (a long sentence).
>
> Going back to the original question regarding KBLaM's generalization to other datasets, we believe the answer is case-by-case
> - If the dataset's is similar to KBLaM's, then KBLaM can be directly generalized. The performance of KBLaM will depend on how similar the dataset is to the synthetic dataset that KBLaM was trained on. Of course, if there is a divergence, the synthetic training dataset should also be updated depending on the intended usecase while applying the same overall methodology. Note that we are introducing KBLaM as a framework for augmenting an LLM with a KB and provide a simple dataset for training such a system rather than providing a final model that can be directly applied to other datasets.
> - If the dataset does not come in the form of triples, then we potentially need to create a separate KB for each dataset where each KB is at least the size of the number of questions we have.
>
> Lastly, it is worth noting that the question of **how** to construct a KB effectively goes beyond the scope of KBLaM, with KBLaM focusing on augmenting a pre-trained LLM with knowledge from an existing KB.
>
> [1] Natural Questions: A Benchmark for Question Answering Research, Kwiatkowski et al., 2019
>
> [4] When Not to Trust Language Models: Investigating Effectiveness of Parametric and Non-Parametric Memories, Mallen et al., 2023
>
> [5] https://huggingface.co/datasets/akariasai/PopQA

---

> ### Comment · Reviewer_6e7T · 2024-11-22
> **Thanks for authors' response**
>
> Thank you to the authors for their prompt response.
>
> With the additional experiments and analysis provided, I now have a clearer understanding of how KBLaM compares to RAG and how well it generalizes to other datasets.
>
> Although KBLaM performs slightly better than RAG, its effectiveness is highly dependent on the datasets used. Furthermore, due to hardware limitations, the evaluation of different KB sizes and model configurations remains constrained.
>
> While the construction of a KB is outside the scope of this paper, it is an essential prerequisite for using KBLaM. Adapting new datasets into a KB format suitable for KBLaM may present another critical and practical challenge.
>
> Nevertheless, I believe KBLaM represents a promising step forward in knowledge augmentation. I have adjusted my scores accordingly.

---

> > ### Author Response · Authors · 2024-11-22
> > **Author response**
> >
> > Thank you for your kind words, your prompt responses and all the informative suggestions. We believe this has lead to a much better, clearer paper.
> >
> > We agree on the current strengths and limitations of KBLaM, which the additional experiments have made clearer to us as well. We are looking forward to these limitations being addressed in future work!

---

### Author Response · Authors · 2024-11-21
**General response regarding KBLaM v.s. RAG**

We would like to thank again to all the reviewers for their helpful and constructive comments!

One common question raised by the reviewers is: *How does KBLaM compare with RAG*?

Before diving into the comparison, we would to emphasize that RAG is a solid and robust method powered by more than 50 years of information retrieval research.
KBLaM on the other hand, "isn't the end-point of this line of research" (pointed out by reviewer 9hRD), but is just *the very first paper* exploring a new possibility of knowledge augmentation.

As such, "KBLaM replaces RAG" is NOT a claim or message of our submission, instead, we aim to answer two fundamental questions:
- Can we augment knowledge as continuous vectors (i.e. a modality) rather than as discrete strings?
- Can we perform retrieval *without an external retriever* but using LLM's attention only?

The answer given by KBLaM is: Yes. Under the settings considered, we successfully encode external knowledge triples as "pre-trained LLM readable" fixed length vectors, and the LLM's attention indeed demonstrates highly interpretable retriever behavior (Fig. 4)

Nevertheless, we agree with reviewer 6e7T on the importance of having a comprehensive comparison between the two approaches, which we analyze from the following four aspects:

**System optimization**

All systems require tuning for optimal performance, same do RAG and KBLaM.

- RAG involves tuning *hyperparameters* such as the chunk size, and the value of K when retrieving the top-K candidates. Notice that these parameters are not trivially optimizable through backpropagation in that: 1. The retrieving process is a discrete operation; 2. The retrievers are external to the LLM and potentially heterogeneous to LLM or non-differentiable.

- KBLaM's "retriever" is essentially the LLM's attention, which can be optimized in a fully differentiable and end-to-end way through backpropagating from any type of LLM's output loss, e.g. cross-entropy loss or RLHF rewards, to the weights.

**Inference latency**

Both RAG and KBLaM require data preprocessing: RAG requires pre-chunking text and storing the index; KBLaM requires pre-computing and caching the key and value.

Assuming the preprocessing is done, given a KB composed of $M$ triples, where each triple takes $T$ tokens, and a user query that is $N$ tokens long, the asymptotic inference time complexity for the two methods is

RAG:
- Retrieve the top-K relevant documents ($O(M)$).
- Tokenize the retrieved documents ($O(KT)$)
- LLM synthesis generation ($O((KT + N)^2)$ from attention, $O(KT + N)$ from FFNs)

KBLaM:
- LLM synthesis generation ($O((M+N)N)$ from attention, $O(N)$ from FFNs.)

*FFN --> Feed-forward networks, also referred to as MLPs*

While RAG's actual latency depends heavily on retriever implementation, we consider an idealized scenario with **highly optimized** retrieval systems and **negligible IO/network/API overhead**, where RAG's overhead only comes from the LLM. Assuming K=5 retrieved triples, our measurements averaged over 40 queries show RAG has "time to first token" of 0.049 seconds, 12% slower than KBLaM's 0.042 seconds with 4,096 triples in the context (full results in Fig.11, appendix).

Interestingly, KBLaM's latency *stays almost constant* across KB sizes. We believe this is because high-FLOP operations in an LLM have input sizes *constant* with respect to KB size in KBLaM:

1. KBLaM's QKV transformation overhead is constant with regard to KB size: All the keys and values for the KB are pre-computed (as we discussed earlier in the paragraph), unlike RAG, which requires computing keys and values for the $K$ triples on the fly.

2. KBLaM's knowledge tokens (i.e. the KB parts) are **not** passed through the FFNs, so the overhead of FFNs is fixed, unlike RAG, whose input to the FFNs scales with the value of $K$.

**Memory overhead**

Similiar to the inference latency, RAG's memory overhead depended on the value of $K$, so it will be fixed as $O((KT + N)^2)$.

KBLaM's memory overhead, $O((M+N)N)$, grows linearly with the KB size.

In our empirical evaluation (Fig.11 in the appendix), we find that given $K=5$, RAG has the same memory overhead as KBLaM with $M=512$.

**Question answering quality**

On the settings we considered, BM25 as a retriever shows almost perfect top-5 accuracy in retrieving relevant triples to the context, therefore RAG would give highly accurate answers similar to in-context learning.

However, in case the question does not have an answer in the KB, RAG's behavior becomes undeterminable: If there is no explicit "unfound detection" mechanism built into the retriever, the pipeline would retrieve wrong triples into the context and LLM may use its own knowledge to synthesize the answer, potentially causing hallucination.
On the other hand, thanks to its favorable memory and latency characteristics, KBLaM can fit the whole KB into the context and is explicitly instruction-tuned to perform refusal when facing such questions.

---

### Author Response · Authors · 2024-12-03
**Final author general response**

# General response

Dear reviewers and AC

We would like to thank again to all the reviewers for their detailed responses and constructive discussion!

To summarize, our submission proposes KBLaM, a new method of augmenting external knowledge represented as a knowledge base (KB) into a pre-trained LLM utilizing the key-value structure in the KB.
KBLaM has the following contributions:
- We show that each triple in the KB can be presented to an LLM as a fixed-length continuous vector rather than discrete tokens.
- We show that LLM's attention can demonstrate an **interpretable** retriever behavior, i.e. it attends accurately over the relevant vector to answer a query (Figure 4 and Figure 5).
- KBLaM allows for augmenting more than 10K triples (~200K tokens of information) into the context of a pre-trained LLM on a single A100 GPU.

The novelty and value of our contributions was recognized by the reviewers as

"...a promising step forward in knowledge augmentation..." (9e7T)

"...this paper points towards a line of research that could be quite fruitful..." (9hRD)

The high-level takeaway message we would like to convey through this work is that:

*If you are attempting to augment a pre-trained LLM with **structured knowledge**, making use of the structure brings you scalability, interpretability, and a principled model that follows the inherent properties of the data.*

KBLaM's design is motivated by the goal of developing a knowledge augmentation strategy that *respects and utilizes* the structure in the knowledge base and the triples, as we discuss in more detail in the introduction section.

---

> ### Author Response · Authors · 2024-12-03
> **Final author general response (2)**
>
> # Summarization of changes
> Below is a summary of the main improvements we made to our work as a result of the reviewer's comments:
>
> - **Reviewer 6e7T** raised the importance of more experiments, comparing KBLaM against other approaches (e.g. RAG), and on other question-answering datasets. We have provided a detailed [comparison](https://openreview.net/forum?id=aLsMzkTej9&noteId=yTL5KMSwTz) in our earlier response, where we show that KBLaM, perhaps surprisingly, has large memory and latency benefits over RAG while being competitive in terms of accuracy. We also conducted additional experiments on a benchmark dataset, PopQA, and we have presented the results in one of our [responses](https://openreview.net/forum?id=aLsMzkTej9&noteId=T1buCAUtw9).
> - **Reviewer q4dg** encouraged us to compare our results with prompt caching ("...for example, by encoding the knowledge base into a key-value cache and reusing it..."), and we have provided a detailed explanation and comparison in terms of the computational overhead introduced by the methods, and the resulting performance in our [response](https://openreview.net/forum?id=aLsMzkTej9&noteId=hE64TYh2yL). They also contributed thoughtful discussion into whether our methods may suffer from over-smoothing with many triples as LLMs regularly do with many tokens, contributing to our understanding of the area. In practice, [we have found this not to be an issue](https://openreview.net/forum?id=aLsMzkTej9&noteId=nFqC0SYI3O).
> *At the end of the discussion phase*, the reviewer suggested a couple of new prompt caching strategies for [comparison](https://openreview.net/forum?id=aLsMzkTej9&noteId=ZUnW9MCnCg): Retrieval + prompt caching + triple caching. While we agree that such methods are relevant, we cannot find existing works that study such regimes and we believe implementing these methods goes beyond the scope of the submission.
> - **Reviewer W1bY** also stressed the importance of more experiments, suggesting baselines such as BM25 which we adopted, and evaluating on more datasets. We believe we have addressed these through our RAG experiments, where we used BM25, and our PopQA results. They also suggested comparisons with structured prompting and attention, which we addressed in [our response](https://openreview.net/forum?id=aLsMzkTej9&noteId=mvsqIwGt5V). Finally, they asked whether we could narrow the gap between KBLaM and ICL results, which we have shown to be possible by using OpenAI's latest embedding model (text-embedding-large) in the appendix.
> - **Reviewer 9hRD** suggested to us an experimental setting where we perturb the queries so there is no exact match in the entity name between the question and the knowledge base (e.g. Asking about 'Maggie Thatcher' rather than 'Margaret Thatcher'). We found that KBLaM continues to retrieve and generate answers accurately in this setting, as illustrated in [this response](https://openreview.net/forum?id=aLsMzkTej9&noteId=jsIpTvEU3H). They also encouraged us to add a theoretical justification for a constant that we use, which we have now clarified in our paper.
> - **Reviewer ouM4** pointed out a few improvements that we have now incorporated, and we have also had discussions over the design and tradeoffs involved in KBLaM. First, we originally only tested the method with one base model, llama 3 8B. We have now reported [results from Phi-3](https://openreview.net/forum?id=aLsMzkTej9&noteId=vdhJEqlMUN), where we found comparable results in terms of retrieval accuracy, but a performance gap in answer quality, potentially because the base model was half the size. Second, as with other reviewers, they suggested a comparison with a RAG baseline. Third, they suggested measurement of the computational overhead of KBLaM, which we have included in our [comparisons with RAG](https://openreview.net/forum?id=aLsMzkTej9&noteId=yTL5KMSwTz), where we found KBLaM to have lower latency generating sequences with 4,096 triples in the context than in-context-learning with 5 triples in the context. Finally, the reviewer raised the important issue of accessibility to our method, considering that the model depends on access to good sentence embeddings. As we highlighted in [our response](https://openreview.net/forum?id=aLsMzkTej9&noteId=QCRC6qp01I), in this aspect, KBLaM is very accessible, with the embeddings used for this KBLaM, OpenAI's ada v2, is incredibly affordable, with a cost of $0.01 to encode a knowledge base with 10,000 triples.
>
> We would like to thank the reviewers again for helping us improve our paper through clarifications, new experiments, and results.
>
> Best Regards,
> Authors of KBLaM

---

### Meta-Review · Area_Chair_a9oz · 2024-12-25

**Metareview:**

This paper focuses on the open-domain question-answering task and proposes the Knowledge Base (KB) Augmented Language Model (KBLaM) framework. Specifically, KBLaM transforms the KB  into continuous key-value vectors using pre-trained encoders, then integrating them into the LLM through an attention mechanism. Unlike Retrieval-Augmented Generation (RAG), the proposed KBLaM does not require retrieval, and, unlike in-context learning (ICL), KBLaM achieves comparable performance with significantly lower computational overhead. Experiments on both a synthesized KB and a real KB dataset demonstrate the effectiveness of the proposed method.

Reviewers generally liked the proposed methods both in motivation and in its potential advantages  (6e7T, q4dg, W1bY, 9hRD, ouM4), and selecting an illuminating set of experiments (6e7T, q4dg). They raised questions about baselines, especially RAG baselines (6e7T, W1bY, ouM4) and that evaluation task on synthetic data / Enron is not the most standard or competitive (6e7T, ouM4) or that evaluating on 1 model is not sufficient (ouM4). Overall the authors addressed most reviewer concerns, in particular convincing comparisons with RAG. The authors also addressed several concerns from the main reject reviewer (ouM4) about resource consumption.

Overall the paper should be accepted. For a method paper, addressing the remaining concerns of reviewer ouM4 could include significant trade-offs in the choice of experiments. Still, for a stronger paper, it could benefit from some experiments on standard competitive tasks (6e7T, ouM4).

**Additional Comments On Reviewer Discussion:**

N/A main reject reviewer did not want to respond, but his review was clear enough and still taken at full weight, so I just followed the discussion thread.

---

### Decision · Program_Chairs · 2025-01-22

Accept (Poster)